# TREATMENT RULE OPTIMIZATION UNDER COUNTERFACTUAL TEMPORAL POINT PROCESSES WITH LATENT STATES

## ABSTRACT

In high-stakes areas like healthcare, retrospective counterfactual analysis—such as evaluating what might have happened if treatments were administered earlier, later, or differently—is vital for refining treatment strategies. This paper proposes a counterfactual treatment optimization framework using temporal point processes to model outcome event sequences. By sampling potential outcome events under new treatment decision rules, our approach seeks to optimize treatment strategies in a counterfactual setting. To achieve accurate counterfactual evaluation of new decision rules, we explicitly introduce latent states into the modeling of temporal point processes. Our method first infers the latent states and associated noise, followed by counterfactual sampling of outcome events. This approach rigorously addresses the complexities introduced by latent states, effectively removing biases in the evaluation of treatment strategies. By proving the identifiability of model parameters in the presence of these states, we provide theoretical guarantees that enhance the reliability and robustness of the counterfactual analysis. By incorporating latent states and proving identifiability, our framework not only improves the accuracy and robustness of treatment decision rules but also offers actionable insights for optimizing healthcare interventions. This method holds significant potential for improving treatment strategies, particularly in healthcare scenarios where patient symptoms are complex and high-dimensional.

## 1 INTRODUCTION

While online reinforcement learning policies have shown promise in designing treatment strategies for sepsis patients in ICU ( Komorowski et al. (2018)), the direct deployment and testing of new treatment strategies on patients raise practical and ethical concerns. Counterfactual evaluation offers a solution by retrospectively assessing the performance of different treatment policies using existing data, without intervening in ongoing patient care. Retrospective analysis is a safer method and has wide applications, as it allows for evaluating new treatments without posing risks to patients (Bal (2009)).

In this paper, we focus on answering the following *what-if* question:

*Given the observational treatment and outcome trajectories, can we modify specific treatment actions to optimize the outcome in a counterfactual manner?*

These modifications must adhere to predefined medical rules. For instance, if some patients respond well to a particular drug, we might explore increasing the dose for better outcomes. Conversely, for patients who do not respond, we could consider switching to alternative medications. These perturbations must follow medical guidelines to ensure safety and efficacy. Similarly, when developing a healthy exercise habit, any changes to the recommended actions must comply with behavior theory principles, such as gradual progression and sustainability. For example, it is inappropriate to recommend excessive exercise or drastic reductions in food intake, as these do not align with established theories of behavior change and can lead to adverse health effects.

Recently, a counterfactual off-policy evaluation method was developed for the partially observable Markov Decision Process (POMDP) Oberst & Sontag (2019). (Noorbakhsh & Rodriguez, 2022)

extended this method to temporal point processes setting. In this context, given a realization of a temporal point process with a known intensity function, a counterfactual sampling algorithm was developed to simulate counterfactual realizations of temporal point processes under a specified alternative intensity function. This developed counterfactual temporal point can be deployed in counterfactual treatment evaluation settings. For instance, it can be utilized to assess the counterfactual treatment effect by sampling outcome events in *what-if* scenarios, where the occurrence of outcome events is modeled by the temporal point processes whose intensity function depends on treatment events.

We aim to extend existing counterfactual off-policy evaluation methods to a treatment decision rule optimization setting, where outcome events are modeled using marked temporal point processes. By integrating counterfactual reasoning with this modeling approach, our method enables the assessment and optimization of treatment strategies in complex, high-dimensional healthcare environments. The framework is composed of two key components: an outer loop for optimizing treatment decision rules and an inner loop for evaluating counterfactual treatment effects.

The outer loop systematically explores the treatment space to identify potential improvements in decision rules. Concurrently, the inner loop evaluates these rules by retrospectively sampling symptom events under counterfactual scenarios. To address the challenges posed by latent states, we introduce a two-stage procedure. Initially, we infer latent states and associated noise to mitigate biases in the marked temporal point processes data, ensuring that our analysis is both accurate and reliable. Importantly, we theoretically prove the identifiability of model parameters in the presence of latent states, providing strong guarantees that enhance the robustness of our counterfactual evaluation. Following this, we conduct counterfactual sampling to rigorously assess the effects of different treatment strategies. This comprehensive approach not only refines existing treatment strategies but also generates new insights for optimizing patient outcomes in healthcare applications.

## 2 RELATED WORK

**Latent states and latent confounders.** The causal inference literature often make the assumption that there are no unobserved confounders (Aglietti et al. (2021); Bica et al. (2021); Vanderschueren et al. (2023)). However, in many practical settings, the NUC assumption could hardly hold. Also, the confounders actually play a crucial role in the counterfactual reasoning process, since we might get a biased result if we ignore the impact of potential confounders on our target variables (Pearl (2009)). In a longitudinal setting, there are several ways to consider the unobserved confounders. One might replace the potential unobserved confounders by some proxies (Louizos et al. (2017); Madras et al. (2019); Kuzmanovic et al. (2021)), or learn substitutes for hidden confounders using some factor models (Bica et al. (2020a); Hatt & Feuerriegel (2024)). In our work, we construct a categorical variable for representing latent states, which can be seen as a partial representation of the unobserved confounders and thus helps mitigate the potential influence (Bartolucci et al. (2022)). Some related works also incorporate a categorical variable to represent latent states in Hawkes processes setting.Xu & Zha (2017) consider a mixture model of Hawkes processes at the sequence level, while Yang & Zha (2013) consider a setting for which the intensity has a mixed kernel. Our setting provides a different view by considering the switching systems represented by the categorical variable.

**Counterfactual reasoning.** Counterfactual reasoning has recently piqued interest in many explainable machine learning works. Note that different from the definition for counterfactual outcome in another line of works (Lim (2018), Melnychuk et al. (2022), Bica et al. (2020b), Hess et al. (2023), Frauen et al. (2024)), here we consider counterfactual works condition on all observed information. To illustrate, they typically refer a conditional average potential outcome related with probability $P(Y_{t+\tau}[a_{[t,t+\tau]}]|\mathcal{H}_t)$, while our counterfactual objective is related to $P(Y_{[0,T]}[a_{[0,T]}]|\mathcal{H}_T)$. The main difference comes from the latter one would condition on the whole observed trajectory, thus we need consider the posterior noise for a SCM. A discrete-time setup, such as POMDP, is considered in many existing works (Oberst & Sontag (2019); Tsirtsis et al. (2021);Aalen et al. (2020);Abid et al. (2022);Tsirtsis & Rodriguez (2024)). The Gumbel-max SCM, a class of SCMs that meets the counterfactual stability criteria for producing counterfactual trajectories in finite POMDPs, is presented by Oberst & Sontag (2019). Noorbakhsh & Rodriguez (2022) apply this special SCM on the thinning process of temporal point process, allowing simulated counterfactual realizations in

continuous time under a given alternative intensity function. This method regards the Lewis' thinning algorithm (Lewis & Shedler (1979)) as the generative method. Therefore, it necessitates the knowledge of an upper bound for both the observed and counterfactual intensity, which is challenging to get when the intensity is history-dependent. To overcome this limitation, Hızlı et al. (2023) extend the counterfactual sampling algorithm to history-dependent point processes by regarding the Ogata's thinning algorithm (Ogata (1981)) as the generative process. However, all these works focus on the univariate case, while we extend the related SCM to the multivariate case, and take the latent states into consideration. Many existing works focus on finding the optimal actions in counterfactual settings. Under static setting, works like (Karimi et al. (2021); Karimi et al. (2020)) focus on finding the actions that one could achieve a better outcome, which belongs to the framework called algorithmic recourse. As for time-varying settings, (Tsirtsis et al. (2021); Tsirtsis & Rodriguez (2024)) provide several methods based on POMDPs which are suitable for different state types in order to find optimal action sequences, but their setting focuses on discrete-time setting. We focus on optimizing the specific meta-rules in a continuous time setting instead of the specific optimal action sequence for an individual, which would be more informative and suitable for flexible situations.

# 3 PROBLEM STATEMENT

## 3.1 OUTCOME AND TREATMENT EVENTS USING HAWKES PROCESS

We utilize a marked temporal point process (MTPP) to model treatment and outcome events, as it provides a natural framework for representing discrete events occurring in continuous time. Specifically, we leverage a multivariate temporal point process, a subclass of MTPPs where event types are represented as distinct dimensions. Within this framework, the Hawkes process (Hawkes (1971)) model the likelihood of future events for each component based on the entire historical sequence across all components. This feature enables the Hawkes process to flexibly capture temporal dependencies and interactions, offering an interpretable structure that is valuable in healthcare settings (Alaa et al. (2017), Nie & Zhao (2022), Bao et al. (2017)).

**Outcome Events:** Let $\{t_{o,j}\}_{j=1}^{N_o}$ denote the times at which outcome events occur, with $N_o$ being the total number of outcome events. Let $\{m_{o,j}\}_{j=1}^{N_o}$ represent the marks (or types) of these outcome events, where $m_{o,j} \in \mathcal{M}$ and $\mathcal{M}$ is the set of outcome event markers. Therefore, the outcome event sequence can be represented as $\{(t_{o,j}, m_{o,j})\}_{j=1}^{N_o}$.

**Outcome Event History:** Denote the history of outcome events up to time $t$ as $\mathcal{H}_o(t)$, which includes all outcome events that have occurred up to time $t$, i.e.,

$$\mathcal{H}_o(t) = \{(t_{o,j}, m_{o,j}) \mid t_{o,j} \leq t\} \tag{1}$$

**Treatment Events and History**: Similarly, we can represent the treatment events as $\{(t_{a,j}, m_{a,j})\}_{j=1}^{N_a}$ where $m_{a,j} \in \mathcal{A}$ and $\mathcal{A}$ is the set of treatment event markers. Denote the history of treatment events up to time $t$ as $\mathcal{H}_a(t)$, i.e.,

$$\mathcal{H}_a(t) = \{(t_{a,j}, m_{a,j}) \mid t_{a,j} \leq t\} \tag{2}$$

**Latent States**: In healthcare settings, for example, the latent states might be the doctors' experience levels and patients' health stages, which are crucial in influencing treatment and outcome events. This paper considers *discrete* and *contemporaneous* latent states, representing $K$ latent factors. We introduce a time-dependent latent variable $\boldsymbol{z}(t) = [z_k]_{k=1,\ldots,K}, \forall t \geq 0$, a one-hot vector indicating which latent factor is active at time $t$. The distribution of $\boldsymbol{z}(t)$ is denoted as $\boldsymbol{\pi} \in \Delta^{K-1}$, which is a probability simplex. By incorporating the latent states, we model the intensity functions of the outcome and treatment events, respectively, as

$$\begin{cases} \boldsymbol{\lambda}_o(t \mid \boldsymbol{z}(t), \mathcal{H}_o(t), \mathcal{H}_a(t)) = \boldsymbol{z}(t)^\top \left( \boldsymbol{\mu}_o + \int_0^t \boldsymbol{\phi}_{o \leftarrow o}(t-s) d\boldsymbol{N}_o(s) + \int_0^t \boldsymbol{\phi}_{o \leftarrow a}(t-s) d\boldsymbol{N}_a(s) \right) \\ \boldsymbol{\lambda}_a(t \mid \boldsymbol{z}(t), \mathcal{H}_o(t), \mathcal{H}_a(t)) = \boldsymbol{z}(t)^\top \left( \boldsymbol{\mu}_a + \int_0^t \boldsymbol{\phi}_{a \leftarrow o}(t-s) d\boldsymbol{N}_o(s) + \int_0^t \boldsymbol{\phi}_{a \leftarrow a}(t-s) d\boldsymbol{N}_a(s) \right) \end{cases} \tag{3}$$

where $\boldsymbol{\lambda}_o(t)$ and $\boldsymbol{\lambda}_a(t)$ are vectors, with each element corresponding to the intensity of a specific type of outcome or treatment event; $\boldsymbol{z}(t)$ selects which component of the intensity function to activate based on the active latent factor; $\boldsymbol{\mu}_o$ and $\boldsymbol{\mu}_a$ are vectors representing the baseline

intensities for outcome and treatment events; $\boldsymbol{N}_o(s)$ and $\boldsymbol{N}_a(s)$ are counting processes, representing the cumulative number of events up to time $s$; the integrals $\int_0^t \boldsymbol{\phi}_{o \leftarrow o}(t-s)d\boldsymbol{N}_o(s)$ and $\int_0^t \boldsymbol{\phi}_{o \leftarrow a}(t-s)d\boldsymbol{N}_a(s)$ represent the contributions of past outcome and treatment events to the current intensity, and $\boldsymbol{\phi}_{o \leftarrow o}(t-s)$ and $\boldsymbol{\phi}_{o \leftarrow a}(t-s)$ are matrices that describe how past events influence the current intensity; similarly, one can interpret the integrals $\int_0^t \boldsymbol{\phi}_{a \leftarrow o}(t-s)d\boldsymbol{N}_o(s)$ and $\int_0^t \boldsymbol{\phi}_{a \leftarrow a}(t-s)d\boldsymbol{N}_a(s)$.

In this paper, among the above integrals, we consider a parametric triggering function $\phi_{m \leftarrow n}(\cdot) : \mathbb{R}^+ \to \mathbb{R}$ of the following form,

$$\phi_{m \leftarrow n}(t) = \beta_{m \leftarrow n}\kappa_{m \leftarrow n}(t) \tag{4}$$

in which the connectivity coefficient $\beta_{m \leftarrow n} \geq 0$ indicates the Granger causal effect from dimension $n$ to $m$, and $\kappa_{m \leftarrow n}(t) : \mathbb{R}^+ \to \mathbb{R}$ is a triggering kernel captures the decay of the dependence on past events. A commonly used example is the exponential transition kernel, $\kappa_{m \leftarrow n}(t) = \exp(-(t))$.
To simplify the notation, from now on, let's denote the conditional intensity function as

$$\lambda_m^*(t \mid \boldsymbol{z}(t)) := \lambda_m(t \mid \boldsymbol{z}(t), \mathcal{H}_o(t), \mathcal{H}_a(t)), \quad \forall m \in \mathcal{M} \cup \mathcal{A} \tag{5}$$

Denote $|\mathcal{M} \cup \mathcal{A}| = U$ and when we use exponential kernel, Eq. 3 could also be written as

$$\lambda_m^*(t \mid \boldsymbol{z}(t)) = \boldsymbol{z}(t)^\top \boldsymbol{\mu}_m + \sum_{n=1}^{U} (\boldsymbol{z}(t)^\top \boldsymbol{\beta}_{m \leftarrow n}) \int_0^{t-} \exp(-(t-s))dN_n(s) \tag{6}$$

where $\boldsymbol{\mu}_m$, and $\boldsymbol{\beta}_{m \leftarrow n}$ are all $K \times 1$ vectors, thus $\boldsymbol{z}(t)^\top(\cdot)$ means choosing one set of parameters according to the current latent state.
We could then conclude our model parameters as $\boldsymbol{\theta} := (\boldsymbol{\pi}, \boldsymbol{\mu}, \boldsymbol{\beta})$, and we will provide sufficient conditions to ensure identifiability in Section 5.

## 3.2 SCM in Ogata's Thinning Process

We assume our treatment and outcome trajectories are generated from Ogata's thinning process (Ogata (1981)). Within a self-defined interval, this process would first sample a potential event with a constant intensity $\lambda_{\text{ub},i}$. The event is then accepted or rejected based on a probability proportional to the ratio of the sum of the target intensities across all dimensions, $\sum_m \lambda_m^*$, to $\lambda_{\text{ub},i}$. This procedure results in two sequences: the observed sequence $\mathcal{H}_{\text{obs}}$, containing accepted events, and the rejected sequence $\mathcal{H}_{\text{rej}}$, containing those that were not accepted. Following ideas in Noorbakhsh & Rodriguez (2022) and Hızlı et al. (2023), we first augment the Ogata's thinning algorithm for MTPP (Algorithm 1) using a structural causal model (SCM) $\mathcal{C}$. We introduce a set of random variables $\boldsymbol{E} \cup \boldsymbol{V} = \{E_1, ..., E_N, V_1, ...V_N\}$, and we assume at time $t_i$, $E_i$ is a binary variable to represent whether $t_i$ is accepted or not, $V_i$ is a categorical variable to represent the mark once $t_i$ is accepted. Therefore, the acceptance and rejection outcomes and the corresponding mark results for the observed sequence $\mathcal{H}_{\text{obs}}$ and the rejected event sequence $\mathcal{H}_{\text{rej}}$, as generated by Ogata's thinning algorithm, can then be encoded through the augmented samples $\{(e_i, v_i)\}_{i=1}^N$, in which we denote $N = |\mathcal{H}_{\text{obs}} \cup \mathcal{H}_{\text{rej}}|$.

Specifically, the SCM $\mathcal{C}$ is defined by the following assignments. Given the latent state $z(t_i)$ at time $t_i$, for $E_i$,

$$E_i = f_E(\lambda_{\text{ub},i}, \boldsymbol{\Lambda}_i, U_i), \quad U_i \sim \text{Unif}(0, \lambda_{\text{ub}}) \tag{7}$$

where $f(\lambda_{\text{ub}}, \boldsymbol{\Lambda}_i, U_i) = \mathbb{I}[U_i \leq \sum_m \Lambda_{i,m}]$, and $\Lambda_{i,m} = \lambda_m^*(t_i \mid z(t_i))$. We could notice that $E_i = 1$ represents the event $t_i$ is accepted, else it is rejected.

Since our setting considers multivariate Hawkes process, we also need to a random variable $V_i$ for the corresponding mark $m_i$ at time $t_i$ once $t_i$ is accepted, i.e., $e_i = 1$,

$$V_i = f_V(E_i, \boldsymbol{\Lambda}_i, \boldsymbol{g}_i), \quad g_{i,j} \sim \text{Gumbel}(0, 1) \tag{8}$$

where $f_V(E_i, \boldsymbol{\Lambda}_i, \boldsymbol{g}_i) = \mathbb{I}_{\{E_i=1\}} \arg\max_j (\log P(Y = j) + g_{i,j})$, $P(Y = j) = \frac{\Lambda_{i,j}}{\sum_m \Lambda_{i,m}}$ and we input $\Lambda_{i,m}$ as same as for $E_i$. From this assignment, we notice only when $t_i$ is accepted we would

have a mark $V_i = m_i$ from the following argmax part, otherwise we would get $V_i = 0$, here we set $0$ as a default value.

Our SCM consists of two types of variables, binary variables $E_i$'s and categorical variables $V_i$'s. We discussed the counterfactual identifiability for this SCM in Appendix C.2.2. Combining these two parts, we would be able to answer the counterfactual questions: what would happened if, at time $t_i$, the intensity had been some different intensity denoted as $\boldsymbol{\lambda}_{\text{cf}}^*(t_i|z(t_i))$ instead of $\boldsymbol{\lambda}_{\text{obs}}^*(t_i|z(t_i))$. Our objective defined in the following section 3.3 actually is in accord with this format.

### 3.3 Objective: Optimizing Treatment in a Counterfactual Manner

Our goal is to answer "what-if" questions: Given the observed sequences of treatment and outcome events, how can we optimize the treatment strategy to improve the final outcome denoted as $Y$ in a counterfactual manner? The final outcome $Y$, such as survival time, is either a direct function of the outcome events or can be directly observed from the outcome events. We assume that $Y$ is measurable given the outcome events.

**Objective:** Instead of optimizing individual treatment actions, we aim to *optimize decision rules* that are pre-specified by doctors. These rules determine the appropriate treatment action based on the patient's condition, reflected in the latent states $z(t)$, and the history of treatment and outcome events. We assume that doctors have prespecified decision rules with fixed conditions but with certain parameters that need to be learned. We can refer to these as **Meta-Rules:**

- **Example Meta-Rule:**
    - **Condition (Fixed)**: If the patient has low blood pressure.
    - **Action (Fixed):** Administer Drug $A$ ($A$ is fixed).
    - **Learnable Parameters:**
        * Dosage: The specific dosage of Drug $A$, denoted as $x$, is learnable.
        * Timing: The best time to administer the drug, $\tau$, is learnable.
        * Latent States Influence: The influence of a latent state $\boldsymbol{z}$, which affects the timing and dosage decision, is learnable.

Given the prespecified meta rule set, denoted as $\{f_d\}_{d \in [D]}$, each meta-rule $f_d\left(x, \tau \mid z_k\right)$ represents the meta-rule $d$ which specifies the treatment action under a given latent state $z_k$. The goal is to optimize $x$ and $\tau$ for each meta-rule corresponding to different latent state $z_k$ to maximize the expected counterfactual outcome $Y$. We formulate the problem as

$$\max_{\{x_{d,k}, \tau_{d,k}\}_{d \in [D], k \in [K]}} \mathbb{E}\left[Y \mid \text{do}\left(\mathcal{H}_a(T) = \mathcal{H}_a'(T) \mid \{f_1, \ldots, f_D\}, \{z_1, \ldots, z_K\}\right), \mathcal{H}_{\text{obs}}(T)\right]$$

$$\text{subject to} \quad x_{d,k} \in [x_{\min}, x_{\max}], \quad \tau_{d,k} \in [\tau_{\min}, \tau_{\max}], \quad \forall d \in [D], k \in [K] \tag{9}$$

We will optimize these decision rules under various patient conditions and histories by adjusting parameters like dosage or timing while keeping the general structure of the rules intact. Here we use $\text{do}\left(\mathcal{H}_a(T) = \mathcal{H}_a'(T) \mid \cdot\right)$ to represent that we revise the treatment trajectories based on the defined meta-rules and the corresponding latent states. Note that this revision would actually result into a revised intensity $\boldsymbol{\lambda}_{\text{cf}}^*(\cdot)$ for outcome events, which means we aim to answer those counterfactual questions as we mentioned in previous part, i.e., perform an *intervention* $\text{do}(\boldsymbol{\Lambda}_i = \boldsymbol{\lambda}_{\text{cf}}(t_i|z(t_i)))$ on both the two SCM $\mathcal{E}_i$ and $\mathcal{V}_i$, given the observed information. To ensure the target outcome is identifiable, we provided causal assumptions we need combined with the counterfactual identifiability of our SCM in Appendix C.

## 4 Model Learning and Inference

To simplify the notation, let's first focus on only one patient's outcome and event data, modeled as a multivariate temporal point process with latent variables, and write down the complete data likelihood. Given the observational treatment and outcome event data $\mathcal{H}(T) := \mathcal{H}_a(T) \cup \mathcal{H}_o(T)$, we aim to jointly learn the model parameters $(\boldsymbol{\pi}, \boldsymbol{\mu}, \boldsymbol{\beta})$ and infer the posterior distribution of $\boldsymbol{z}(t)$ at each time $t \in \{t_j\}$, where $\{t_j\} := \{t_{o,j} \mid t_{o,j} < T\} \cup \{t_{a,j} \mid t_{a,j} < T\}$, which contains all the outcome event time and treatment event time for each this patient.

Given the conditional probability decomposition of

$$P_{\boldsymbol{\mu},\boldsymbol{\beta}}\left(\mathcal{H}(T) \mid \boldsymbol{z}\right) = \prod_j P_{\boldsymbol{\mu},\boldsymbol{\beta}}^*\left((t_j, m_j) \mid \boldsymbol{z}(t_j); \boldsymbol{\mu}, \boldsymbol{\beta}\right) \tag{10}$$

$$= \prod_j \lambda_{m_j}^*(t_j \mid \boldsymbol{z}(t_j), \boldsymbol{\mu}, \boldsymbol{\beta}) \exp\left(-\int_{t_{j-1}}^{t_j} \lambda_{\text{sum}}^*(s \mid \boldsymbol{z}(s); \boldsymbol{\mu}, \boldsymbol{\beta}) ds\right) \tag{11}$$

where $\lambda_{\text{sum}}^* = \sum_{m \in \mathcal{M} \cup \mathcal{A}} \lambda_m^*$ aggregates the intensities over all possible event types. Given the above formula, we can write down the complete-data likelihood as follows:

$$P_{\boldsymbol{\mu},\boldsymbol{\beta}}(\mathcal{H}(T), \boldsymbol{z}) = \prod_j \prod_{k=1}^K \left[\pi_k \cdot P_{\boldsymbol{\mu},\boldsymbol{\beta}}^*((t_j, m_j) \mid z_k(t_j) = 1; \boldsymbol{\mu}, \boldsymbol{\beta})\right]^{\mathbb{1}(z_k(t_j)=1)}. \tag{12}$$

Note that the above formula is the complete data likelihood since we don't know the latent variable. We will adopt EM algorithm to learn the model parameters and infer $\boldsymbol{z}(t)$.

**E-step: Update Responsibility.**  Compute the posterior distribution of latent states at each time $t_j$ given the current parameters:

$$P\left(z(t_j) \mid \mathcal{H}(T), \boldsymbol{\pi}^{\text{old}}, \boldsymbol{\mu}^{\text{old}}, \boldsymbol{\beta}^{\text{old}}\right) \text{ for each patient at each time } t_j$$

The posterior distribution is computed using Bayes' theorem:

$$P\left(z(t_j) \mid \mathcal{H}(T), \boldsymbol{\pi}^{\text{old}}, \boldsymbol{\mu}^{\text{old}}, \boldsymbol{\beta}^{\text{old}}\right) \propto P\left((t_j, m_j) \mid \boldsymbol{z}(t_j), \boldsymbol{\mu}^{\text{old}}, \boldsymbol{\beta}^{\text{old}}\right) P\left(\boldsymbol{z}(t_j)\right)$$

Therefore

$$P\left(z_k(t_j) = 1 \mid \mathcal{H}(T), \boldsymbol{\pi}^{\text{old}}, \boldsymbol{\mu}^{\text{old}}, \boldsymbol{\beta}^{\text{old}}\right) = \frac{\pi_k^{\text{old}} P_{\boldsymbol{\mu}^{\text{old}},\boldsymbol{\beta}^{\text{old}}}^*\left((t_j, m_j) \mid z_k(t_j) = 1\right)}{\sum_{k'=1}^K \pi_{k'}^{\text{old}} P_{\boldsymbol{\mu}^{\text{old}},\boldsymbol{\beta}^{\text{old}}}^*\left((t_j, m_j) \mid z_{k'}(t_j) = 1\right)} \tag{13}$$

We will denote $\gamma_{kj} := P\left(z_k(t_j) = 1 \mid \mathcal{H}(T), \boldsymbol{\pi}^{\text{old}}, \boldsymbol{\mu}^{\text{old}}, \boldsymbol{\beta}^{\text{old}}\right)$.

**M-step: Update Parameters.**

$$\pi_k^{\text{new}} = \frac{n_k}{N_a + N_o}, \quad n_k = \sum_{j=1}^{N_a + N_o} \gamma_{kj}, \quad \forall k \in [K] \tag{14}$$

where $n_k$ is the expected number of times the latent variable is in state $k$. The updates for $\boldsymbol{\mu}$ and $\boldsymbol{\beta}$ involve maximizing the expected complete-data log-likelihood:

$$\boldsymbol{\mu}^{\text{new}}, \boldsymbol{\beta}^{\text{new}} = \arg\max_{\boldsymbol{\mu},\boldsymbol{\beta}} \sum_j \sum_k \gamma_{kj} \log P_{\boldsymbol{\mu},\boldsymbol{\beta}}^*\left((t_j, m_j) \mid z_k(t_j) = 1\right) \tag{15}$$

The above derivation focuses on the event data of a single patient. To generalize this to multiple patients, let $\mathcal{H}^i(T^i)_{i \in [I]}$ represent the event data for all patients, where $i$ is the patient index. We can then extend the EM algorithm to handle these multiple-patient scenarios easily. The complete derivation can be found in Appendix E.

## 5 IDENTIFIABILITY OF MIXTURE MODEL PARAMETERS

We are interested in understanding the conditions our model must satisfy so that the following implication holds for all $(\mathcal{H}, \boldsymbol{z})$ :

$$\forall (\boldsymbol{\theta}, \boldsymbol{\theta}') : \quad P_{\boldsymbol{\theta}}(\mathcal{H}) = P_{\boldsymbol{\theta}'}(\mathcal{H}) \implies \boldsymbol{\theta} = \boldsymbol{\theta}' \tag{16}$$

That is, if any two different sets of model parameters $\boldsymbol{\theta}$ and $\boldsymbol{\theta}'$ result in the same marginal distribution $P_{\boldsymbol{\theta}}(\mathcal{H})$, then this would imply that these parameters are identical, leading to matching joint distributions $P_{\boldsymbol{\theta}}(\mathcal{H}, \boldsymbol{z})$. This implies that if we learn parameters $\boldsymbol{\theta}$ such that $P_{\boldsymbol{\theta}}(\mathcal{H}) = P_{\boldsymbol{\theta}^*}(\mathcal{H})$ (the ideal case where $\boldsymbol{\theta}^*$ represents the true underlying parameters), then the corresponding joint distribution also matches: $P_{\boldsymbol{\theta}}(\mathcal{H}, \boldsymbol{z}) = P_{\boldsymbol{\theta}^*}(\mathcal{H}, \boldsymbol{z})$. If the joint distribution matches, it ensures that we have identified the correct prior $P_{\boldsymbol{\theta}}(\boldsymbol{z}) = P_{\boldsymbol{\theta}^*}(\boldsymbol{z})$ and the correct posteriors $p_{\boldsymbol{\theta}}(\boldsymbol{z} \mid \mathcal{H}) = p_{\boldsymbol{\theta}^*}(\boldsymbol{z} \mid \mathcal{H})$. This guarantees that the EM algorithm, by maximizing the likelihood, correctly identifies the underlying parameters, ensuring the model's identifiability.

**Assumption 1.** *(Bonnet et al. (2023)) We assume that a.s. for every $(i,j) \in \{\mathcal{M} \cup \mathcal{A}\}^2$, $i \neq j$, there exist an event time $\tau$ from counting process $N^j$, and an event time $\tau_+ > \tau$ from process $N^i$, such that:*

1. *$\lim_{t \to \tau^-} \lambda_{i,\{\boldsymbol{\mu}_i, \boldsymbol{\beta}_i\}}(t) > 0$*

2. *there are only events of process $N^j$ in the interval $[\tau, \tau_+)$.*

**Theorem 1.** *Assume that the number of latent factors $K$ is identified using some auxiliary argument. The true categorical distribution $F^0$ of latent states is uniformly identified, and given state $k$, assume each Hawkes system satisfies Assumption 1, the corresponding parameters $\boldsymbol{\mu}_k$ and $\boldsymbol{\beta}_k$ are identifiable, i.e., for any $\{\boldsymbol{\mu}_k', \boldsymbol{\beta}_k'\}$,*

$$\forall i \in \{\mathcal{M} \cup \mathcal{A}\}, \lambda^*_{i,\{\boldsymbol{\mu}_{k,i}, \boldsymbol{\beta}_{k,i}\}}(t) = \lambda^*_{i,\{\boldsymbol{\mu}'_{k,i}, \boldsymbol{\beta}'_{k,i}\}}(t) \ a.e. \iff \{\boldsymbol{\mu}_k, \boldsymbol{\beta}_k\} = \{\boldsymbol{\mu}'_k, \boldsymbol{\beta}'_k\}$$

Assumption 1 requires that one process is not totally inhibited, and that there exists an interval during which only events from this process occur. These conditions are generally reasonable and are likely to be met in most practical scenarios. This section focus on the mixture model parameters $\boldsymbol{\theta}$'s identifiability, which contributes to guarantee the performance of EM algorithm and counterfactual analysis process. The identifiability of the parameters $\boldsymbol{\theta}$ rules out that there exists different parameters that entails the same distribution for the observed data, which ensures that for any revised treatment plan, the corresponding intensity is uniquely determined, enabling a reliable and consistent counterfactual analysis. Note that to guarantee these parameters represents the causal relationships and identifiability for our counterfactual expected outcome, we need additional causal assumptions as we mentioned in previous section and provided in Appendix C.2. We provide detailed proof for model parameters identifiability in Appendix F.

## 6 DECISION RULE OPTIMIZATION ALGORITHM

Given the historical data, we have already applied the EM algorithm to estimate the model parameters and know how to infer the latent states in a closed form. Now given the optimization formulation as shown in Eq. (9), let's specify the decision rule optimization algorithm. In our setting, the dosage can be discretized into different treatment event markers or types.

**Output:** Optimized treatment decision rules $\left\{m^*_{d,k}, \tau^*_{d,k}\right\}_{d \in [D], k \in [K]}$.

**Step 1: Initialization** - Initialize the treatment decision rule parameters $\{m_{d,k}, \tau_{d,k}\}_{d \in [D], k \in [K]}$, where $m_{d,k}$ represents a discretized dosage level (treatment marker) and $\tau_{d,k}$ represents the treatment time.

**Step 2: 1. Outer Loop - Treatment Decision Rule Optimization** Repeat until convergence: For each decision rule parameter $m_{d,k}$ and $\tau_{d,k}$, perform a gradient-based or combinatorial optimization:

$$\{m_{d,k}, \tau_{d,k}\} \leftarrow \{m_{d,k}, \tau_{d,k}\} + \eta \nabla_{\{m_{d,k}, \tau_{d,k}\}} \mathbb{E}\left[Y \mid \text{do}\left(\mathcal{H}'_a(T) \mid \{f_1, \ldots, f_D\}, \{z_1, \ldots, z_K\}, \mathcal{H}_{\text{obs}}(T)\right)\right]$$

Note: $m_{d,k} \in \mathcal{A}_{d,k}$ belongs to a discrete set, which is a subset of $\mathcal{A}$ defined in the meta rule and $\tau_{d,k}$ belongs to a continuous set.

Here we provide the policy gradient method we used for learning the optimal policy for both treatment type and time. The detailed gradient estimation method and description could be found in Appendix D.

- Discrete Treatment Marker $m_{d,k}$: we represent the selection of a discrete treatment marker $m_{d,k}$ using a probability vector $\mathbf{p}_{d,k}$ where each element $p^{(i)}_{d,k}$ represents the probability of selecting the $i$ th marker. Then we can use the softmax function directly:

$$\mathbf{p}_{d,k} = \text{Softmax}\left(\mathbf{s}_{d,k}\right)$$

where $\mathbf{s}_{d,k}$ are the logits (unconstrained parameters).

- Continuous Treatment Time $\tau_{d,k}$: here $\tau_{d,k}$ represents the time lag for performing this treatment once the condition is satisfied. We parameterize the continuous treatment time with Gaussian kernel,

$$\pi(\tau_{d,k}) = \frac{1}{\sigma\sqrt{2\pi}}\exp\left(-\frac{(\nu_{d,k} - \tau_{d,k})^2}{2\sigma^2}\right)$$

By fixing the variance $\sigma^2$ as a small value, optimizing the mean $\nu_{d,k}$ would equivalently guide to the best choice of treatment time.

**2. Inner Loop - Counterfactual Treatment Effect Evaluation:** Evaluate the effectiveness of the current decision rule by sequentially sampling the outcome events under counterfactual scenarios.

1. Latent State Inference (EM Algorithm): Use the same approach as before to infer the posterior probability of latent states $z$ using the EM algorithm.

2. Counterfactual Sampling: The CF algorithm mainly consists of two parts for sampling counterfactual outcomes, the detailed Algorithm 3 is presented in Appendix C.3.
   - Sample from Posterior of Latent States and Noise: Sample latent states $z$ from the posterior and noise $u$ in acceptance-rejection parts, get the counterfactual outcome event intensity function.
   - Generate Counterfactual Outcomes: Simulating symptom events with the inferred latent states and sampled noise.

3. Evaluate Treatment Effects: Assess $Y$ using the counterfactual outcomes and update the decision rule parameters accordingly.

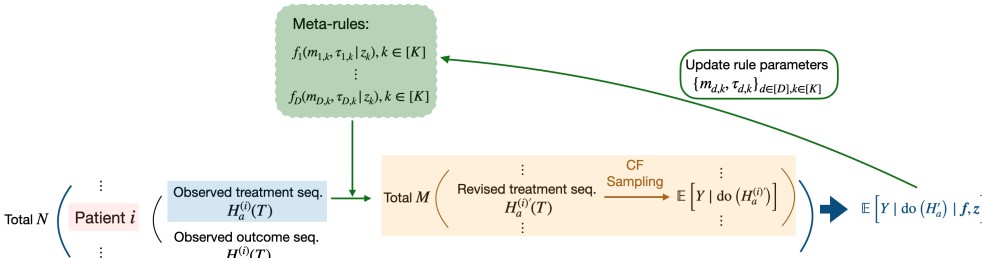

Figure 1: Decision-rule optimization framework as described in Section 6.

# 7 EXPERIMENTS

## 7.1 SYNTHETIC EXPERIMENT

**Experimental setup.** To validate our method, we constructed an 8-dimensional Hawkes process with four dimensions representing treatments ($A_1$, $A_2$, $B_1$, $B_2$) and four as outcomes. Indicators 1 and 2 reflect worsening symptoms, while indicators 3 and 4 reflect improvement. Drug $A$ targets indicator 1, and drug $B$ targets indicator 2, with $A_1/B_1$ representing lower dosages and $A_2/B_2$ higher dosages. Treatments also increase the likelihood of positive outcomes (indicators 3 and 4). We incorporated two latent states to represent patient health stages, with healthier states having lower probabilities of adverse events. Intensity parameters were designed to reflect these relationships.

To evaluate patient outcomes, we defined a deterministic outcome $Y$, calculated as the square of the weighted proportion of positive outcomes, with later events receiving higher weights. Our goal was to optimize decision rules that maximize $Y$, focusing on two meta-rules detailed in Appendix G.1.

We first generate 600 sequences from the ground truth model as we described above. Then based on these sequences, we learn the model parameters by our EM methods and denoted as model 1. We also applied original MLE method and have model 2 which does not take latent states into consideration. For optimizing our meta-rules, we simulate a synthetic baseline population dataset. This

baseline dataset is constructed by only retaining the outcome intensity parts from model 1 for simulating the outcomes and adopting some naive policies for choosing potential improper treatments when some outcomes occurs, e.g., when outcome 1 occurs at state 0 we choose drug $A_1$ instead of drug $A_2$, thus obviously they are not the best policies. By performing our decision-rule optimization algorithm, we could then compare our current policies performance with the baseline performance.

**Results.** We want to compare the optimization results from our model 1 with latent states and model 2 without latent states. Model 2 (without latent states) converges faster, as shown in Fig. 2 (a), due to its simpler structure. However, this comes at the cost of reduced accuracy in learning true preferences. Model 1 (with latent states) achieves higher counterfactual rewards and accurately learns the ground truth rule-type preferences, while Model 2 struggles to capture these due to its lack of latent state representation.

To evaluate the impact of learned meta-rules, we applied them to a synthetic data simulator, comparing results against baseline rules and optimized rules without latent states. Each approach generated 500 sequences. As shown in Fig. 2 (b), optimized rules incorporating latent states consistently achieved higher expected rewards, despite similar ranges of variation. This highlights how latent state-based models effectively capture hidden dynamics or unobservable patient conditions, offering more precise and adaptive recommendations compared to non-latent and baseline models.

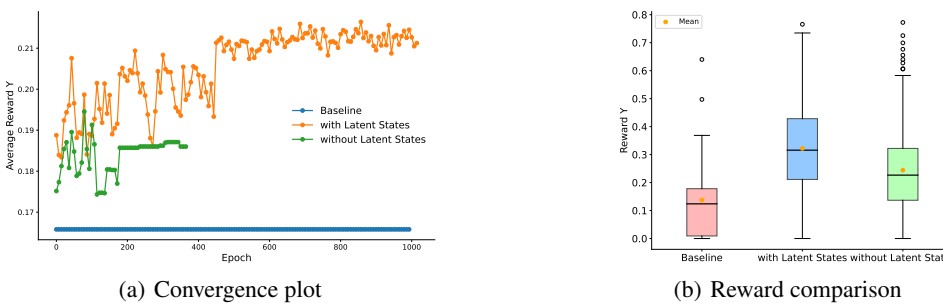

(a) Convergence plot

(b) Reward comparison

Figure 2: Synthetic experiment results. **(a)** the convergence performance during the optimization process for models with or without latent states. **(b)** the box-plot for comparing reward from baseline rules and the two types of optimized rules.

### 7.2 EXPERIMENTS ON REAL-WORLD DATA

Sepsis, a life-threatening condition caused by the body's overactive response to infection, leads to inflammation, tissue damage, organ failure, and high mortality rates. Despite advances in critical care, clinical recommendations for sepsis management remain uncertain, highlighting the need for decision-rule optimization techniques like ours (Evans et al. (2021)). To address this challenge, we utilized the MIMIC-III database (Johnson et al. (2016)), a widely used resource containing de-identified health data from over 60,000 ICU patients. While MIMIC-III supports predictive modeling and treatment evaluation, the common *no unobserved confounders* assumption is difficult to meet, as unrecorded factors or omitted variables can influence outcomes. Our approach, designed to account for latent states, leverages this database to mitigate confounding influences and optimize decision rules for sepsis management.

We extracted 2,000 patient sequences meeting the criteria for sepsis diagnosis (Saria (2018)). These patients formed the population for our EM algorithm so as to fit our mixture model. We then select the patients based on our meta-rules, ensuring those sequences containing the potential treatment action to be revised, and we use this subset for our decision-rule optimization process. Treatments for sepsis typically involve vasopressor therapy and fluid administration, with the aim of stabilizing patients by maintaining blood pressure and ensuring proper organ perfusion (Komorowski et al. (2018)). For outcomes, we monitored real-time urine output and survival, key indicators in sepsis management. Low urine output is often an early sign of kidney dysfunction and septic shock, potentially signaling inadequate treatment response or impending multi-organ failure. Ultimately, improving survival rates is the overarching goal of any sepsis intervention. We detailed these treat-

ments and outcomes in Table 2. The reward design and specific meta-rules we aim to optimize is described also in Appendix G.2.

**Results.** We hypothesized the existence of two latent states in the data and used the EM algorithm to estimate parameters. Latent State 1, associated with stable conditions, showed lower baseline event rates and inter-event influence, suggesting minimal need for intervention. In contrast, Latent State 2 reflected acute conditions with higher event rates and stronger inter-event influence, requiring more proactive care. In both states, predefined triggers like low urine output and low blood pressure effectively prompted appropriate treatments, validating our rules.

Analyzing optimized meta-rules from the MIMIC-III dataset revealed state-dependent treatment patterns. For fluids, stable conditions led to administration 0.6560 time units after low urine detection, compared to 0.7994 in acute cases. Vasopressors were administered earlier in acute states (0.6027 vs. 0.8433 time units after low blood pressure). Preferences for crystalloid fluids in stable states shifted toward colloids in acute ones, while vasopressor usage balanced between norepinephrine and dopamine in severe cases. Feedback from ChatGPT 4.0 confirmed the clinical validity of these meta-rules, emphasizing their utility in distinguishing and managing patient conditions effectively.

| Latent State | Meta-Rule | Event Distribution | Time |
|---|---|---|---|
| Latent State 1 | Rule 1 | Colloid 0.27, Crystalloid 0.45, Water 0.29 | 0.656 |
| | Rule 2 | Norepinephrine 0.42, Dopamine 0.58 | 0.799 |
| Latent State 2 | Rule 1 | Colloid 0.37, Crystalloid 0.39, Water 0.24 | 0.843 |
| | Rule 2 | Norepinephrine 0.46, Dopamine 0.54 | 0.603 |

Table 1: Probability distribution for different latent states and meta-rules. The y-axis represents the probability, while the x-axis represents events.

## 8 CONCLUSIONS

We introduced a counterfactual treatment optimization framework leveraging temporal point processes to model treatment-outcome event sequences while addressing challenges posed by latent states. This framework provides insights into optimizing treatment strategies in complex healthcare settings, enhancing clinical decision-making and patient outcomes. Future work could improve its practical applicability by developing methods to automatically determine the optimal number of latent states and extending the framework to model time-dependent latent states, capturing delayed or evolving influences throughout the treatment course.

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

## A    OGATA THINNING ALGORITHM FOR MULTIVARIATE TPP

---

**Algorithm 1** Modified Ogata's Thinning Algorithm of MTPP

---

**Input**      : $t_o$, $T$, $\lambda(t, m_i)(i = 1, ...M)$, interval function $l(t)$
**Initialize:** $t = 0, \mathcal{H} = \emptyset$
1 **Function** OGATA $(t_0, T, l, \lambda)$ **:**
2 $\quad$ $t = t_0,$
3 $\quad$ **while** $t < T$ **do**
4 $\quad\quad$ $\lambda_{\max}(t) = \max\limits_{t' \in (t, t+l(t))} (\sum_{i=1}^{M} \lambda(t', m_i))$
5 $\quad\quad$ $u_0 \sim U(0, 1),$
6 $\quad\quad$ $\Delta t = -(\ln u_0)/\lambda_{\max}.$
7 $\quad\quad$ **if** $\Delta t < l(t)$ **then**
8 $\quad\quad\quad$ **if** $u_a < \frac{\sum_{i=1}^{M} \lambda(t+\Delta t, m_i)}{\lambda_{\max}}, u_a \sim U(0, 1)$ **then**
9 $\quad\quad\quad\quad$ Draw event type $m \sim \frac{\lambda(t+\Delta t, m)}{\sum_{i=1}^{M} \lambda(t+\Delta t, m_i)}$
10 $\quad\quad\quad\quad$ $\mathcal{H} = \mathcal{H} \cup (t + \Delta t, m)$
11 $\quad\quad\quad$ **end**
12 $\quad\quad\quad$ $t = t + \Delta t$
13 $\quad\quad$ **else**
14 $\quad\quad\quad$ $t = t + l(t)$
15 $\quad\quad$ **end**
16 $\quad$ **end**
17 **return** $\mathcal{H}$

---

## B    GUMBEL-MAX TRICK FOR SAMPLING COUNTERFACTUAL MARK

### B.1    DRAW SAMPLES FROM GIVEN CATEGORY DISTRIBUTION

If the logits for discrete random variables $X_1, X_2, ..., X_K$ are $\theta_1, \theta_2, ..., \theta_K$, we can use the softmax function to define the sampling probability $\pi_i$ of $X_i$:

$$\pi_i = \frac{\exp\{\theta_k\}}{\sum_{k=1}^{K} \exp\{\theta_k\}}$$

Meanwhile, we can also use Gumbel trick (Huijben et al. (2022)) to achieve the same result, which is equivalent to adding the standard gumbel noise $g_k$ to the log-likelihood and take argmax of it. Denote $\alpha = \exp(\theta)$, the distribution is the same as using softmax function

$$\arg\max_{k \in 1, ..., K}(\log \alpha_k + g_k) \sim \frac{\alpha_k}{\sum_{k=1}^{K} \alpha_k}, \quad g_k \sim \text{Gumbel}(0, 1)$$

### B.2    POSTERIOR DISTRIBUTION OF GUMBEL NOISE FROM GIVEN SAMPLES

Suppose a variable $X$ has a categorical distribution and we already observe the outcome $X_k$, we can also recover the posterior Gumbel noise that produces the result (Maddison & Tarlow (2017)).

Denote $Z = \sum_{k=1}^{K} \alpha_k$, the maximum value is distributed as a standard Gumbel

$$\max_{k \in 1, ..., K}(\log \alpha_k + g_k) \sim \text{Gumbel}(\log Z)$$

If we observe the outcome is $k$, then the posterior probability for $g_i$, $i \neq k$ is:

$$p(g_i|k, g_k) = \frac{f_{\log \alpha_i}(g_i)[g_k \geq g_i]}{F_{\log \alpha_i}(g_k)}$$

where $f_{\log \alpha_i}$ and $F_{\log \alpha_i}$ represent the PDF and CDF of a Gumbel with location $\log \alpha_i$ respectively, and $[A]$ is the Iverson bracket notation: $[A] = 1$ if $A$ is True, otherwise $[A] = 0$. This means the remaining Gumbels are independent Gumbels with location truncated at $g_k$.

And for the Gumbel variable $g_k$ (where $X_k$ is the chosen variable) is:

$$p(g_k) = f_{\log Z}(g_k)$$

which means $g_k$ is distributed as a Gumbel with location $\log Z$.

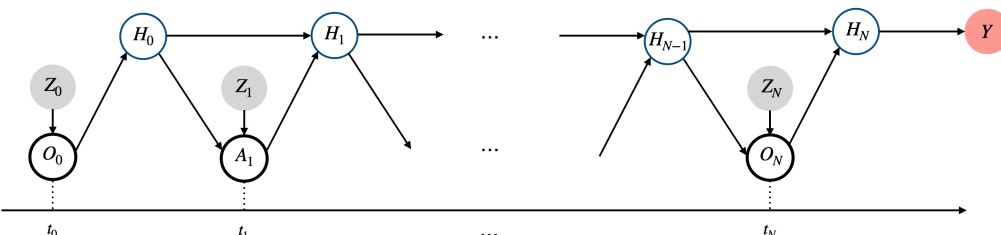

Figure 3: Our framework consider a sequential treatment-outcome setup in continuous time. The latent state $Z_t$ and the history $H_{t-1}$ containing all past treatments and outcomes would have effects on the treatment or outcome event occur at time $t$.

## C COUNTERFACTUAL TRAJECTORIES

### C.1 POSTERIOR DISTRIBUTION FOR NOISE

Here we provide detailed posterior distribution for two types of SCMs at a time event $t_i$ we constructed in section 3.2.

**Posterior for $U_i$**   In the assignment for $E_i$, we have an independent noise $U_i \sim \text{Unif}(0, \lambda_u b)$. Therefore we could easily get the posterior given the latent state $\boldsymbol{z}(t_i)$,

$$p(U_i \mid t_i, \lambda_{\text{ub},i}, \boldsymbol{\lambda}^*_{\text{obs}}, \boldsymbol{z}(t_i)) = \begin{cases} \text{Unif}(0, \sum_m \lambda^*_{\text{obs},m}(t_i \mid \boldsymbol{z}(t_i))), & \text{if } t_i \text{ is observed,} \\ \text{Unif}(\sum_m \lambda^*_{\text{obs},m}(t_i \mid \boldsymbol{z}(t_i)), \lambda_{\text{ub},i}), & \text{if } t_i \text{ is rejected.} \end{cases}$$

**Posterior for $g_i$**   In the assignment for $V_i$, once we have $E_i = 1$, the argmax part is equivalent to a Gumbel-max SCM. Based on B.1 and B.2, considering the mark of an observed event follows a categorical distribution, we could get the corresponding posterior Gumbel noise. Thus when we performing counterfactual sampling process with those observed events, we should sample their mark with following algorithm:

---

**Algorithm 2** Counterfactual Mark Sampling

---

**Input**     : $\lambda_{\text{obs}}(t, m_i), \lambda_{\text{cf}}(t, m_i), m_{\text{obs}}$
**Initialize:** $G \sim \text{Gumbel}(0, 1), \alpha_j = \frac{\lambda_{obs}(t, k_j)}{\sum_j \lambda_{obs}(t, k_j)}, \alpha'_j = \frac{\lambda_{cf}(t, k_j)}{\sum_j \lambda_{cf}(t, k_j)}$
18 **Function** CFmark_sample ($\lambda_{\text{obs}}(t, m_i), \lambda_{\text{cf}}(t, m_i), m_{\text{obs}}$)**:**
19   | **if** $m_i == m_{\text{obs}}$ **then**
20   |   | $g_j = G - \log(\alpha_j)$
21   | **else**
22   |   | $g_j = \text{TruncatedGumbel}(\log(\alpha_j), G) - \log(\alpha_j)$
23   | **end**
24   | $m = \underset{m' \in 1, \dots, M}{\arg\max} (\log \alpha_{m'} + g_{m'})$
25 **return** $m$

---

In the above algorithm, the truncated Gumbel is defined as

$$\text{TruncatedGumbel}(\log(\alpha_i), G) = -\log(\exp(-G - \log(\alpha_i)) + \exp(-G - \log(\sum_i \alpha_i)))$$

## C.2 IDENTIFIABILITY OF THE COUNTERFACTUAL OBJECTIVE FUNCTION

Our objective function is a counterfactual outcome,

$$\mathbb{E}\left[Y \mid \mathrm{do}\left(\mathcal{H}_a(T) = \mathcal{H}'_a(T) \mid \{f_1, \ldots, f_D\}, \{z_1, \ldots, z_K\}\right), \mathcal{H}_{\mathrm{obs}}(T)\right]$$

For simplicity, in this section we will denote treatment and outcome events in a time interval $[t, t + \tau]$ as $\boldsymbol{A}_{[t,t+\tau]}$ and $\boldsymbol{O}_{[t,t+\tau]}$. Since we regard the target outcome $Y$ as a deterministic function of outcome trajectories, i.e., $Y = g(\boldsymbol{O}_{[0,T]})$, this computation is related to the following counterfactual distribution for outcomes:

$$P(\boldsymbol{O}_{[0,T]}[\boldsymbol{A}_{[0,T]} = \mathcal{H}'_a(T)] \mid \mathcal{H}_{\mathrm{obs}}(T))$$

To ensure the identifiability of our counterfactual objective function, we first need some standard causal assumptions to ensure this distribution could be answered by our model, and we also need to guarantee the counterfatual result of the defined SCM are identifiable.

### C.2.1 CAUSAL ASSUMPTIONS

We make the following assumptions,

**Assumption 2.** *(Consistency) given a sequence of treatment events $\boldsymbol{A}_{[t,t+\tau]} = \boldsymbol{a}_{[t,t+\tau]}$, $t \geq 0$ and $\tau \in [0, \Delta]$, the potential outcome events $\boldsymbol{O}_{[t,t+\tau]}[\boldsymbol{a}_{[t,t+\tau]}]$ coincides with the observed outcome $\boldsymbol{a}_{[t,t+\tau]}$.*

**Assumption 3.** *(Continous-Time Positivity) Given any history $\mathcal{H}_{<t}$, there is a positive probability of receiving treatment at any point $t$, all possible treatment mark $m$ and all possible latent states $\boldsymbol{z}$, i.e., the conditional treatment intensity satisfies $0 < \lambda^*_{m \in \mathcal{A}}(t \mid \boldsymbol{z}) < 1$.*

**Assumption 4.** *(Relaxed continuous-time NUC) Condition on the past history and latent states, the conditional treatment intensity is independent of the potential outcome trajectories, i.e., $\lambda^*_{m \in \mathcal{A}}(t \mid \boldsymbol{z}_t) = \lambda^*_{m \in \mathcal{A}}(t \mid \boldsymbol{z}_t, \mathcal{F}(\boldsymbol{O}_s[a'_{(t,s)}] : s > t))$, where $\mathcal{F}(\boldsymbol{O}_s[a'_{(t,s)}] : s > t))$ is the filtration generated by future potential outcomes.*

Similar to the G-computation formula in (Robins (1986)), we factorize $\boldsymbol{O}_{[0,T]}$ in time-order (assume total $N$ events here without loss of generality),

$$P(\boldsymbol{O}_{[0,T]}[\boldsymbol{a}_{[0,T]}] \mid \mathcal{H}_{\mathrm{obs}}(T)) = \prod_{i=1}^{N} P(\boldsymbol{O}_{t_i}[\boldsymbol{a}_{[t_{i-1},t_i)}] \mid \boldsymbol{O}_{[0,t_{i-1}]}, \boldsymbol{a}_{[0,t_{i-1})}, \mathcal{H}_{\mathrm{obs}}(T))$$

$$= \prod_{i=1}^{N} P(\boldsymbol{O}_{t_i}[\boldsymbol{a}_{[t_{i-1},t_i)}] \mid \mathcal{H}_{\mathrm{cf},t_{i-1}}, \mathcal{H}_{\mathrm{obs}}(T))$$

Focus on the probability at time $t_i$, based on the above assumptions we have,

$$P(\boldsymbol{O}_{t_i}[\boldsymbol{a}_{[t_{i-1},t_i)}] \mid \mathcal{H}_{\mathrm{cf},t_{i-1}}, \mathcal{H}_{\mathrm{obs}}(T)) = P(\boldsymbol{O}_{t_i} \mid \mathcal{H}_{\mathrm{cf},t_{i-1}}, \mathcal{H}_{\mathrm{obs}}(T)) \tag{A.1}$$

$$= P(\boldsymbol{O}_{t_i} \mid \mathcal{H}_{\mathrm{cf},t_{i-1}}, \boldsymbol{a}_{[t_{i-1},t_i)}, \boldsymbol{z}_{[t_{i-1},t_i]}, \mathcal{H}_{\mathrm{obs}}(T)) \tag{A.3}$$

Our model combined with counterfactual sampling algorithm we introduced in Section C.3 would entail this distribution.

### C.2.2 IDENTIFIABILITY OF COUNTERFACTUALS

For the binary variable $E_i$, we would state the assignment for it satisfies the monotonicity condition, which is a sufficient assumption to identify binary counterfatuals.

**Definition 1.** *(Monotonicity, Pearl (2000)) An SCM $\mathcal{E}$ of a binary variable $Y$ is monotonic with respect to a binary variable $T$ if and only if the condition,*

$$\mathbb{E}[Y = y \mid \mathrm{do}(T = t')] \geq \mathbb{E}[Y = y \mid \mathrm{do}(T = t)]$$

*implies that $P(Y = y' \mid Y = y, T = t, \mathrm{do}(T = t')) = 0$, where $y' \neq y$.*

**Proposition 1.** *Let binary variable $T = \{\boldsymbol{\lambda}_{\mathrm{cf}}(t_i | \boldsymbol{z}(t_i)), \boldsymbol{\lambda}_{\mathrm{obs}}(t_i | \boldsymbol{z}(t_i))\}$, our SCM of thinning for time point $t_i$ satisfies monotonicity condition.*

*Proof:* For a given time point $t_i$, we could have the probability of accepting it under an interventional distribution over $\mathcal{E}$,

$$P(E_i = 1 \mid \mathrm{do}(\boldsymbol{\Lambda}_i = \boldsymbol{\lambda}(t_i))) = P(E_i = 1 \mid \boldsymbol{\Lambda}_i = \boldsymbol{\lambda}(t_i)) = \frac{\sum_m \lambda_m(t_i)}{\lambda_{\mathrm{ub}}}$$

Then similar as the proof from Hızlı et al. (2023), we mainly consider the two cases of the input point $t_i$,

- Suppose we observe $E_i = 0$, i.e., $t_i$ is a rejected point in $\mathcal{H}_{\mathrm{rej}}$, then if we perform an intervention to decrease the summation of intensity over all dimensions as $\sum_m \lambda^*_{\mathrm{cf},m}(t_i \mid \boldsymbol{z}(t_i)) \leq \sum_m \lambda^*_{\mathrm{obs},m}(t_i \mid \boldsymbol{z}(t_i))$, we have

$$\mathbb{E}[E_i = 0 \mid \mathrm{do}(\boldsymbol{\Lambda}_i = \boldsymbol{\lambda}_{\mathrm{cf}}(t_i \mid \boldsymbol{z}(t_i)))] \geq \mathbb{E}[E_i = 0 \mid \mathrm{do}(\boldsymbol{\Lambda}_i = \boldsymbol{\lambda}_{\mathrm{obs}}(t_i \mid \boldsymbol{z}(t_i)))]$$
$$\implies P(E_i = 1 \mid E_i = 0, \boldsymbol{\Lambda}_i = \boldsymbol{\lambda}_{\mathrm{obs}}(t_i \mid \boldsymbol{z}(t_i)), \mathrm{do}(\boldsymbol{\Lambda}_i = \boldsymbol{\lambda}_{\mathrm{cf}}(t_i \mid \boldsymbol{z}(t_i)))) = 0$$

  follows from we have the posterior distribution $U_{\mathrm{rej}} \sim \mathrm{Unif}(\sum_m \lambda^*_{\mathrm{obs},m}(t_i \mid \boldsymbol{z}(t_i)), \lambda_{\mathrm{ub}})$, and we have $\sum_m \lambda^*_{\mathrm{cf},m}(t_i \mid \boldsymbol{z}(t_i)) \leq \sum_m \lambda^*_{\mathrm{obs},m}(t_i \mid \boldsymbol{z}(t_i))$, thus we would get $U_{\mathrm{rej}} \geq \sum_m \lambda^*_{\mathrm{cf},m}(t_i \mid \boldsymbol{z}(t_i))$ and reject this point $t_i$ again.

- Suppose we observe $E_i = 1$, i.e., $t_i$ is an accepted point in $\mathcal{H}_{\mathrm{obs}}$, then if we perform an intervention to increase the summation of intensity over all dimensions as $\sum_m \lambda^*_{\mathrm{cf},m}(t_i \mid \boldsymbol{z}(t_i)) \leq \sum_m \lambda^*_{\mathrm{obs},m}(t_i \mid \boldsymbol{z}(t_i))$, we have

$$\mathbb{E}[E_i = 1 \mid \mathrm{do}(\boldsymbol{\Lambda}_i = \boldsymbol{\lambda}_{\mathrm{cf}}(t_i \mid \boldsymbol{z}(t_i)))] \geq \mathbb{E}[E_i = 1 \mid \mathrm{do}(\boldsymbol{\Lambda}_i = \boldsymbol{\lambda}_{\mathrm{obs}}(t_i \mid \boldsymbol{z}(t_i)))]$$
$$\implies P(E_i = 0 \mid E_i = 1, \boldsymbol{\Lambda}_i = \boldsymbol{\lambda}_{\mathrm{obs}}(t_i \mid \boldsymbol{z}(t_i)), \mathrm{do}(\boldsymbol{\Lambda}_i = \boldsymbol{\lambda}_{\mathrm{cf}}(t_i \mid \boldsymbol{z}(t_i)))) = 0$$

  follows from we have the posterior distribution $U_{\mathrm{obs}} \sim \mathrm{Unif}(0, \sum_m \lambda^*_{\mathrm{obs},m}(t_i \mid \boldsymbol{z}(t_i)))$, and we have $\sum_m \lambda^*_{\mathrm{cf},m}(t_i \mid \boldsymbol{z}(t_i)) \geq \sum_m \lambda^*_{\mathrm{obs},m}(t_i \mid \boldsymbol{z}(t_i))$, thus we would get $U_{\mathrm{obs}} \leq \sum_m \lambda^*_{\mathrm{cf},m}(t_i \mid \boldsymbol{z}(t_i))$ and accept this point $t_i$ again.

For the categorical variable $V_i$, we note that it depends on the value of $E_i$, and the remaining part would be a Gumbel-max trick,

- Suppose we observe $E_i = 1$, i.e., $t_i$ is an accepted point in $\mathcal{H}_{\mathrm{obs}}$, then $V_i = m_i$ as the observed mark $m_i$. Decided by the counterfactual value of $E_i$, we would have following two cases,
  - If $E_{\mathrm{cf},i} = 0$, then $V_{\mathrm{cf},i} = 0$.
  - If $E_{\mathrm{cf},i} = 1$, in this case, the assignment for $V_i$ could be regarded as a Gumbel-max SCM. Following result from Oberst & Sontag (2019), this part satisfies the counterfactual stability condition and thus the counterfactuals of $V_i$ would be identifiable.
- Suppose we observe $E_i = 0$, i.e., $t_i$ is a rejected point in $\mathcal{H}_{\mathrm{rej}}$, then $V_i = 0$, and we do not have the posterior information about the Gumbel-max part. Similarly, we would have following two cases,
  - If $E_{\mathrm{cf},i} = 0$, then $V_{\mathrm{cf},i} = 0$.
  - If $E_{\mathrm{cf},i} = 1$, in this case, we need to perform the Gumbel-max part directly since we do not have the prior knowledge from observed data, and this would not violate the counterfatual identifiability.

Therefore, we conclude that our counterfactual query is identifiable.

### C.3 Counterfactual Sampling Algorithm for MTPP with latent state

Our counterfactual sampling algorithm for multivariate TPP with latent state is derived from Hızlı et al. (2023), which based on Ogata thinning algorithm as mentioned in Appendix A.

About the choice of interval function $l(\tau)$ in the algorithm, we in practice choose the one would returns the next observed event after time $\tau$, which means the observation period $[0, T]$ would be split into intervals with end points $(0, t_1, ..., t_N, T)$. In counterfactual process, for the interval $[t_i, t_{i+1}]$,

i.e., the two adjacent observed event times, the prior probability for choosing the the latent state $z$ would be $\gamma^{\mathrm{obs}}_{t_{i+1}}$, which is the posterior we calculated in E-step from observation sequence. The latent state would also be affected by the previous counterfactual results $\mathcal{H}^{\mathrm{cf}}_{<t_{i+1}}$, and suppose the previous event in counterfactual results before $t_{i+1}$ is $(\tau^{\mathrm{cf}}, m^{\mathrm{cf}}_\tau)$ thus we could have the following posterior probability for interval $[\tau^{\mathrm{cf}}, t_{i+1}]$,

$$
\begin{aligned}
\gamma_{k,[\tau^{\mathrm{cf}},t_{i+1}]} &:= P\left(z_k(\tau^{\mathrm{cf}}) = 1 \mid \mathcal{H}^{\mathrm{cf}}_{<t_{i+1}}, \gamma^{\mathrm{obs}}_{t_{i+1}}\right) \\
&= \frac{\gamma^{\mathrm{obs}}_{k,t_{i+1}} P\left((\tau^{\mathrm{cf}}, m^{\mathrm{cf}}_\tau) \mid z_k(\tau^{\mathrm{cf}}) = 1, \mathcal{H}^{\mathrm{cf}}_{<t_{i+1}}\right)}{\sum_{k'=1}^{K} \gamma^{\mathrm{obs}}_{k',t_{i+1}} P\left((\tau^{\mathrm{cf}}, m^{\mathrm{cf}}_\tau) \mid z_{k'}(\tau^{\mathrm{cf}}) = 1, \mathcal{H}^{\mathrm{cf}}_{<t_{i+1}}\right)}
\end{aligned}
\tag{17}
$$

The conditional probability is,

$$
P\left((\tau^{\mathrm{cf}}, m^{\mathrm{cf}}_\tau) \mid z_k(\tau^{\mathrm{cf}}) = 1, \mathcal{H}^{\mathrm{cf}}_{<t_{i+1}}\right) = \lambda_{m^{\mathrm{cf}}_\tau}(\tau^{\mathrm{cf}} \mid \boldsymbol{\theta}^k, \mathcal{H}^{\mathrm{cf}}_{<t_{i+1}}) \exp\left(-\int_{t^{\mathrm{cf}}_{<\tau}}^{\tau^{\mathrm{cf}}} \lambda_{\mathrm{sum}}(s \mid \boldsymbol{\theta}^k, \mathcal{H}^{\mathrm{cf}}_{<t_{i+1}})ds\right)
$$

in which $t^{\mathrm{cf}}_{<\tau}$ represents the previous event time in current counterfactual results before $\tau^{\mathrm{cf}}$. We then sample the latent state for interval $[\tau^{\mathrm{cf}}, t_{i+1}]$ based on the above $\boldsymbol{\gamma}_{[\tau^{\mathrm{cf}},t_{i+1}]}$.

Based on the current $z$, we could easily calculate the maximum intensity $\lambda_{\mathrm{ub}}$ in this interval. Then we first sample the potential rejected point in the interval and decide the counterfactual acceptance result of this point. If the generated rejected point falls out the interval, we would then discard this point and consider the endpoint $t_{\mathrm{obs},i}$ instead.

---

**Algorithm 3** Counterfactual Sampling Algorithm For MTPP with latent state

---

**Input** : $T, \mathcal{H}_{\text{obs}}, \boldsymbol{\gamma}^{\text{obs}}$, interval function $l(\cdot), \lambda_{\text{obs}}(t, m_i), \lambda_{\text{cf}}(t, m_i)(i = 1, ..., M)$

**Output:** Counterfactual results $\mathcal{H}_{\text{cf}} = \{\mathbf{o}_i = (t_i, m_i)\}_{i=1}^{N_{\text{cf}}}$

26 **Function** CFSAMPLE $(T, l, \lambda_{\text{obs}}, \lambda_{\text{cf}}, \mathcal{H}_{\text{obs}})$ :

27     $\tau = 0, \mathcal{H}_{\text{cf}} = \emptyset$

      **while** $\tau < T$ **do**

28         $\boldsymbol{\gamma}_{[\tau, \tau+l(\tau)]} = \{\frac{\gamma^{\text{obs}}_{k, \tau+l(\tau)} P((\tau, m_\tau)|z_k(\tau)=1, \mathcal{H}_{\text{cf}})}{\sum_{k'=1}^{K} \gamma^{\text{obs}}_{k', \tau+l(\tau)} P((\tau, m_\tau)|z_{k'}(\tau)=1, \mathcal{H}_{\text{cf}})}\}_{k=1}^{K}$

29         $\boldsymbol{z} \sim \text{Categorical}(\boldsymbol{\gamma}_{[\tau, \tau+l(\tau)]})$,

30         $\lambda_{\text{ub}} = \sup_{s \in [\tau, \tau+l(\tau)]} \{\lambda^*(s) : \lambda^* \in \{\sum_{i=1}^{M} \lambda_{\text{obs}}(s, m_i|\boldsymbol{z}), \sum_{i=1}^{M} \lambda_{\text{cf}}(s, m_i|\boldsymbol{z})\}\}$.

31         $t_{\text{rej}} = \text{OGATA}(\tau, \tau + l(\tau), l, \lambda_{\text{ub}}, \lambda_{\text{ub}} - \sum_{i=1}^{M} \lambda_{\text{obs}}(t, m_i|\boldsymbol{z}))$.

32         **if** $t_{\text{rej}} < l(\tau)$ and $t_{\text{rej}} + \tau \leq T$ **then**

33           $u_{\text{rej}} \sim U(\sum_{i=1}^{M} \lambda_{\text{obs}}(\tau + t_{\text{rej}}, m_i|\mathcal{H}_{\text{obs}}, \boldsymbol{z}), \lambda_{\text{ub}})$.

34           **if** $u_{\text{rej}} \leq \sum_{i=1}^{M} \lambda_{\text{cf}}(\tau + t_{\text{rej}}, m_i|\mathcal{H}_{\text{cf}}, \boldsymbol{z})$ **then**

35             $m \sim \frac{\lambda_{\text{cf}}(\tau+t_{\text{rej}}, m|\mathcal{H}_{\text{cf}}, \boldsymbol{z})}{\sum_{i=1}^{M} \lambda_{\text{cf}}(\tau+t_{\text{rej}}, m_i|\mathcal{H}_{\text{cf}}, \boldsymbol{z})}$,

36             $\mathcal{H}_{\text{cf}} = \mathcal{H}_{\text{cf}} \cup (\tau + t_{\text{rej}}, m)$,

37           **end**

38           $\tau = \tau + t_{\text{rej}}$.

39         **else**

40           **if** $\tau + l(\tau) \in \mathcal{H}_{\text{obs}}$ **then**

41             $t_{\text{obs}} = \tau + l(\tau)$,

42             $m_{\text{obs}} = \mathcal{H}_{\text{obs}}[t_{\text{obs}}][1]$,

43             **if** $m_{\text{obs}} \notin \mathcal{A}$ **then**

44               $u_{\text{obs}} \sim U(0, \sum_{i=1}^{M} \lambda_{\text{obs}}(t_{\text{obs}}, m_i|\mathcal{H}_{\text{obs}}, \boldsymbol{z}))$.

45               **if** $u_{\text{obs}} \leq \sum_{i=1}^{M} \lambda_{\text{cf}}(t_{obs}, m_i|\mathcal{H}_{\text{cf}}, \boldsymbol{z})$ **then**

46                 $m = \text{CFmark\_sample}(\lambda_{\text{obs}}(t_{\text{obs}}, m_i|\mathcal{H}_{\text{obs}}, \boldsymbol{z}),$

                   $\lambda_{\text{cf}}(t_{\text{obs}}, m_i|\mathcal{H}_{\text{cf}}, \boldsymbol{z}), m_{\text{obs}})$,

47                 $\mathcal{H}_{\text{cf}} = \mathcal{H}_{\text{cf}} \cup (t_{\text{obs}}, m)$,

48               **end**

49             **end**

50           **end**

51           $\tau = \tau + l(\tau)$.

52         **end**

53       **end**

54     **return** $\mathcal{H}_{\text{cf}}$

---

## D   DECISION RULE OPTIMIZATION

In each meta-rule at a specific latent state $f_d(m_{d,k}, \tau_{d,k}|z_k)$, we assume we our decision following some probabilistic policies. Here we use $\pi_{\theta_{\tau_{d,k}}}$ and $\pi_{\theta_{m_{d,k}}}$ to represent counterfactual policies for treatment time and treatment type.

- Discrete Treatment Marker $m_{d,k}$: we represent the selection of a discrete treatment marker $m_{d,k}$ using a probability vector $\mathbf{p}_{d,k}$ where each element $p_{d,k}^{(i)}$ represents the probability of selecting the $i$ th marker. Then we can use the softmax function directly:

$$\mathbf{p}_{d,k} = \text{Softmax}(\mathbf{s}_{d,k})$$

where $\mathbf{s}_{d,k}$ are the logits (unconstrained parameters). Thus the probability for choosing $i$-th marker defined in the meta-rule is,

$$\pi(m_{d,k} = i) = p_{d,k}^{(i)} = \frac{\exp(s_{d,k}^{(i)})}{\sum_{j=1}^{J} \exp(s_{d,k}^{(j)})}$$

- Continuous Treatment Time $\tau_{d,k}$: here $\tau_{d,k}$ represents the time lag for performing this treatment once the condition is satisfied. We parameterize the continuous treatment time with Gaussian kernel,

$$\pi(\tau_{d,k}) = \frac{1}{\sigma\sqrt{2\pi}} \exp\left(-\frac{(\nu_{d,k} - \tau_{d,k})^2}{2\sigma^2}\right)$$

By fixing the variance $\sigma^2$ as a small value, optimizing the mean $\nu_{d,k}$ would give us enough information about the best choice of treatment time,

We estimate the gradient with respect to the parameters $\mathbf{s}_{d,k}$ and $\nu_{d,k}$ by score function estimators when optimizing the meta-rules. In the following we give the detailed gradient with respect to a specific patient $j$'s trajectory.

**Derivative with respect to $\nu_{d,k}$.** For $\nu_{d,k}$, we first get the gradient of the log-likelihood of revised action sequence for a specific patient.

$$\nabla_{\nu_{d,k}} \log(p(\mathcal{H}_a^{(j)\prime}; \boldsymbol{\nu}, \boldsymbol{s})) = \nabla_{\nu_{d,k}}(-\frac{1}{2}\sum_i \frac{(\nu_{d,k} - \tau_{d,k,i})^2}{\sigma^2}) = \frac{\sum_i(\tau_{d,k,i} - \nu_{d,k})}{\sigma^2} \tag{18}$$

in which the summation over $i$ means we need to sum over all the revised treatments triggered by the meta-rule $f_d(m_{d,k}, \tau_{d,k}|z_k)$. Then we can get the score function estimator for the gradient with respect to $\nu_{d,k}$:

$$\nabla_{\nu_{d,k}} \mathbb{E}_{p(\mathcal{H}_a^{(j)\prime}; \boldsymbol{\nu}, \boldsymbol{s})}[\mathbb{E}[Y|do(\mathcal{H}_a^{(j)} = \mathcal{H}_a^{(j)\prime})]] = \mathbb{E}_{p(Y, \mathcal{H}_a^{(j)\prime}; \boldsymbol{\nu}, \boldsymbol{s})}[\nabla_{\nu_{d,k}} \log(p(\mathcal{H}_a^{(j)\prime}; \boldsymbol{\nu}, \boldsymbol{s}))Y]$$

**Derivative with respect to $\mathbf{s}_{d,k}$.** The parts containing $\mathbf{s}_{d,k}$ of the log-likelihood of revised action sequence for a specific patient would be,

$$\log(p(\mathcal{H}_a^{(j)\prime}; \boldsymbol{\nu}, \boldsymbol{s})) = \sum_i \log(\frac{\exp(s_{d,k}^{(m_{d,k,i})})}{\sum_{j=1}^J \exp(s_{d,k}^{(j)})}) = \sum_i(s_{d,k}^{(m_{d,k,i})} - \log(\sum_{j=1}^J \exp(s_{d,k}^{(j)})))$$

here we also use the summation over $i$ to represent all the revised treatments triggered by the meta-rule $f_d(m_{d,k}, \tau_{d,k}|z_k)$. For $\mathbf{s}_{d,k}$, we could then get the gradient of this log-likelihood of revised action sequence for a specific patient,

$$\nabla_{\mathbf{s}_{d,k}} \log(p(\mathcal{H}_a^{(j)\prime}; \boldsymbol{\nu}, \boldsymbol{s})) = \{(-\frac{\sum_i \exp(s_{d,k}^{(j)})}{\sum_{j'=1}^J \exp(s_{d,k}^{(j')})} + \sum_i \mathbb{I}\{m_{d,k,i} = j\})\}_{j=1}^J \tag{19}$$

Then we can get the score function estimator for the gradient with respect to $\mathbf{s}_{d,k}$:

$$\nabla_{\mathbf{s}_{d,k}} \mathbb{E}_{p(\mathcal{H}_a^{(j)\prime}; \boldsymbol{\nu}, \boldsymbol{s})}[\mathbb{E}[Y|do(\mathcal{H}_a(T) = \mathcal{H}_a^{(j)\prime})]] = \mathbb{E}_{p(Y, \mathcal{H}_a^{(j)\prime}; \boldsymbol{\nu}, \boldsymbol{s})}[\nabla_{\mathbf{s}_{d,k}} \log(p(\mathcal{H}_a^{(j)\prime}; \boldsymbol{\nu}, \boldsymbol{s}))Y)]$$

## E  THE DETAILS OF EM ALGORITHM

### E.1  EM FOR ONE SEQUENCE

Here we focus on the single patient first. The complete-data log-likelihood is

$$\ell(\boldsymbol{\mu}, \boldsymbol{\beta}; \mathcal{H}, \boldsymbol{z}) = \sum_{j=1}^N \sum_{k=1}^K \mathbb{1}(z_k(t_j) = 1) \left[\log \pi_k + \log P_{\boldsymbol{\mu}, \boldsymbol{\beta}}^*((t_j, m_j) \mid z_k(t_j) = 1; \boldsymbol{\mu}, \boldsymbol{\beta})\right]. \tag{20}$$

where

$$P_{\boldsymbol{\mu}, \boldsymbol{\beta}}^*((t_j, m_j) \mid z_k(t_j) = 1; \boldsymbol{\mu}, \boldsymbol{\beta}) = \lambda_{m_j}^*(t_j \mid z_k(t_j) = 1; \boldsymbol{\mu}, \boldsymbol{\beta}) \exp\left(-\int_{t_{j-1}}^{t_j} \lambda_{\text{sum}}^*(s \mid z_k(s) = 1; \boldsymbol{\mu}, \boldsymbol{\beta})ds\right) \tag{21}$$

For our mixed model, at a event $(t_j, m_j)$, we have

$$P(\boldsymbol{z}(t_j)) = \prod_{k=1}^{K} \pi_k^{z_{kj}}, \quad P((t_j, m_j) \mid \boldsymbol{z}(t_j)) = \prod_{k=1}^{K} \left( \lambda_{m_j}^*(t_j \mid \boldsymbol{\theta}^k) \exp\left( -\int_{t_{j-1}}^{t_j} \lambda_{\text{sum}}^*(s \mid \boldsymbol{\theta}^k) ds \right) \right)^{z_{kj}}$$

Suppose our time period is $[0, T]$, $\{\boldsymbol{z}(t_j)\}_{j=1}^N$ and $\boldsymbol{z}(T)$ for period $[t_N, T]$ are given, the likelihood for the complete data is,

$$L_C(\boldsymbol{\pi}, \boldsymbol{\theta}) = [\prod_{j=1}^{N} \prod_{k=1}^{K} \pi_k^{z_{kj}} \left( \lambda_{m_j}(t_j | \boldsymbol{\theta}^k) \exp\left( -\sum_{u'=1}^{U} \int_{t_{j-1}}^{t_j} \lambda_{u'}(s | \boldsymbol{\theta}^k) ds \right) \right)^{z_{kj}} ]$$

$$\times \prod_{k=1}^{K} \exp\left( -\sum_{u'=1}^{U} \int_{t_N}^{T} \lambda_{u'}(s | \boldsymbol{\theta}^k) ds \right)^{z_{kT}} \tag{22}$$

The corresponding log-likelihood is,

$$l_C(\boldsymbol{\pi}, \boldsymbol{\theta}) = \sum_{j=1}^{N} \sum_{k=1}^{K} z_{kj} \left( \log \pi_k + \log(\lambda_{m_j}(t_j | \boldsymbol{\theta}^k)) - \sum_{u'=1}^{U} \int_{t_{j-1}}^{t_j} \lambda_{u'}(s | \boldsymbol{\theta}^k) ds \right)$$

$$- \sum_{k=1}^{K} z_{kT} \left( \sum_{u'=1}^{U} \int_{t_N}^{T} \lambda_{u'}(s | \boldsymbol{\theta}^k) ds \right) \tag{23}$$

**E-step: Update Responsibility**  The detailed E-step for a single patient is described in Section 4.

**M-step: Update Parameters**  By maximizing the expectation of the complete log-likelihood of the complete data we could then estimate the parameters of Hawkes process.

$$\mathbb{E}_{\boldsymbol{z}}(l_C(\boldsymbol{\pi}, \boldsymbol{\theta})) = \sum_{j=1}^{N} \sum_{k=1}^{K} \gamma_{kj}(\log \pi_k + \log(\lambda_{m_j}(t_j | \boldsymbol{\theta}^k)) - \sum_{u'=1}^{U} \int_{t_{j-1}}^{t_j} \lambda_{u'}(s | \boldsymbol{\theta}^k) ds)$$

$$- \sum_{k=1}^{K} \gamma_{kT} \left( \sum_{u'=1}^{U} \int_{t_N}^{T} \lambda_{u'}(s | \boldsymbol{\theta}^k) ds \right)$$

$$= \sum_{j=1}^{N} \sum_{k=1}^{K} \gamma_{kj}(\log \pi_k + \log(\mu_{m_j}^k + \sum_{n<j} \beta_{m_j \leftarrow m_n}^k \kappa(t_j - t_n))$$

$$- \sum_{u'=1}^{U} ((t_j - t_{j-1})\mu_{u'}^k + \sum_{l=1}^{j-1} (\beta_{u' \leftarrow m_l}^k \int_{t_{j-1}}^{t_j} \kappa(s - t_l) ds)))$$

$$- \sum_{k=1}^{K} \gamma_{kT} \left( \sum_{u'=1}^{U} ((T - t_N)\mu_{u'}^k + \sum_{l=1}^{N} (\beta_{u' \leftarrow m_l}^k \int_{t_N}^{T} \kappa(s - t_l) ds))) \right) \tag{24}$$

For exponential kernel, the integral in the last term could be further written as $\int_{t_{j-1}}^{t_j} \kappa(s - t_l) ds = \exp(-(t_{j-1} - t_l)) - \exp(-(t_j - t_l))$.

One can directly perform gradient descent on Eq. 24 for solving $\boldsymbol{\theta}^{\text{new}}$, and update for $\pi_k$ by,

$$\pi_k^{\text{new}} = \frac{n_k}{N_a + N_o}, \quad n_k = \sum_{j=1}^{N_a + N_o} \gamma_{kj}, \quad \forall k \in [K]$$

### E.2  EM FOR MULTIPLE SEQUENCES

Based on the above single patient case, we could easily generalize our EM algorithm to multiple sequences case. Consider we have a set of sequences denoted as $\mathcal{H} = \{\mathcal{H}^i(T)\}_{i \in [I]}$, we assume all

sequences share same $\boldsymbol{\pi}$ and $\boldsymbol{\theta}$, and all the time of events are in $[0, T]$. We then get our complete likelihood over all sequences as follows,

$$L(\boldsymbol{\pi}, \boldsymbol{\theta}) = \prod_{i \in [I]} P(\mathcal{H}^i(T) \mid \boldsymbol{z}) = \prod_{i \in [I]} L_C^i(\boldsymbol{\pi}, \boldsymbol{\theta}) \tag{25}$$

where $L_C^i(\boldsymbol{\pi}, \boldsymbol{\theta})$ is the complete likelihood for sequence $\mathcal{H}^i(T)$, as we described in Eq. 22.

**E-step: Update Responsibility**    Similarly as single sequence case, given the current parameters, we compute the posterior distribution of latent states at each time $t_{i,j}$, here $t_{i,j}$ is the $j$-th event in $i$-th sequence:

$$P\left(z_k(t_{i,j}) = 1 \mid \mathcal{H}^i(T), \boldsymbol{\pi}^{\text{old}}, \boldsymbol{\theta}^{\text{old}}\right) = \frac{\pi_k^{\text{old}} P_{\boldsymbol{\mu}^{\text{old}}, \boldsymbol{\theta}^{\text{old}}}^* \left((t_{i,j}, m_{i,j}) \mid z_k(t_{i,j}) = 1\right)}{\sum_{k'=1}^K \pi_{k'}^{\text{old}} P_{\boldsymbol{\mu}^{\text{old}}, \boldsymbol{\theta}^{\text{old}}}^* \left((t_{i,j}, m_{i,j}) \mid z_{k'}(t_{i,j}) = 1\right)} \tag{26}$$

We will denote $\gamma_{ikj} := P\left(z_k(t_{i,j}) = 1 \mid \mathcal{H}^i(T), \boldsymbol{\pi}^{\text{old}}, \boldsymbol{\theta}^{\text{old}}\right)$.

**M-step: Update parameters**    We could then have our expected complete-data log-likelihood for all sequences,

$$\mathbb{E}_{\boldsymbol{z}}(l_C(\boldsymbol{\pi}, \boldsymbol{\theta})) = \sum_{i \in [I]} \mathbb{E}_{\boldsymbol{z}}(l_C^i(\boldsymbol{\pi}, \boldsymbol{\theta})) \tag{27}$$

$$= \sum_{i \in [I]} \left(\sum_{j=1}^{N_i} \sum_{k=1}^K \gamma_{ikj}(\log \pi_k + \log(\mu_{m_{i,j}}^k + \sum_{n<j} \beta_{m_{i,j} \leftarrow m_{i,n}}^k \kappa(t_{i,j} - t_{i,n})) \right.$$

$$- \sum_{u'=1}^U ((t_{i,j} - t_{i,j-1})\mu_{u'}^k + \sum_{l=1}^{j-1} (\beta_{u' \leftarrow m_{i,l}}^k \int_{t_{i,j-1}}^{t_{i,j}} \kappa(s - t_{i,l})ds)))$$

$$\left. - \sum_{k=1}^K \gamma_{ikT} (\sum_{u'=1}^U ((T - t_{N_i})\mu_{u'}^k + \sum_{l=1}^{N_i} (\beta_{u' \leftarrow m_{i,l}}^k \int_{t_{N_i}}^T \kappa(s - t_{i,l})ds))))) \right) \tag{28}$$

Then we update for $\boldsymbol{\theta}$ by maximizing Eq. 28:

$$\boldsymbol{\theta}^{\text{new}} = \arg\max_{\boldsymbol{\theta}} \quad l(\boldsymbol{\pi}, \boldsymbol{\theta}) \tag{29}$$

One can also solve this by gradient descent, on similarly could get the closed-form answer for the surrogate function as in the single sequence case. And for updating $\boldsymbol{\pi}$, similarly we have,

$$\pi_k^{\text{new}} = \frac{\sum_{i \in [I]} n_{i,k}}{\sum_{i \in [I]} (N_{i,a} + N_{i,o})}, \quad n_{i,k} = \sum_{j=1}^{N_{i,a}+N_{i,o}} \gamma_{ikj}, \quad \forall k \in [K] \tag{30}$$

where $n_{i,k}$ is the expected number of times the latent variable is in state $k$ in the $i$-th sequence.

# F    THE PROOF OF IDENTIFIABILITY

**Proof for Theorem 1: Uniformly Identifiability**    Our proof mainly consists of two parts, (1) given the latent state $z_k = 1$, the parameters $\boldsymbol{\theta}^k$ in the Hawkes intensity are identifiable, (2) we prove that the distribution of categorical variable $\boldsymbol{z}$ is uniformly identified.

Firstly, suppose the latent state is given, i.e., $z_k = 1$, then the corresponding parameters $\boldsymbol{\mu}_k$ and $\boldsymbol{\beta}_k$ are identifiable follows the identifiability result of multivariate Hawkes processes under certain conditions, as described in Theorem 3.1 in Bonnet et al. (2023) and we assume the required assumptions are satisfied.

Secondly, we prove the distribution of our categorical distribution is uniformly identified.

Let $\Theta = \{\boldsymbol{\theta}^k\}_{k=1}^K$ be the support of the random coefficients of the intensity, $\mathcal{F}(\Theta)$ be the set of all distributions on that support, $\mathcal{X}$ be the support of the covariates $\boldsymbol{x}$, and $F^0$ be the true distribution. We then introduce the definition of *uniformly identified* here,

**Definition 2.** *The distribution $F^0 \in \mathcal{F}(\Theta)$ is uniformly identified over choices of $(\Theta, \mathcal{X}_0)$ if for any $F^1 \in \mathcal{F}(\Theta)$, $F^1 \neq F^0$, there exists $\mathcal{X}_0^1 \subset \mathcal{X}_0$ such that $P(\boldsymbol{x}, F^0) - P(\boldsymbol{x}, F^1) \neq 0$ for all $\boldsymbol{x} \in \mathcal{X}_0^1$ for any choice of the support of random coefficients $\Theta$ and the subset of the support of covariates $\mathcal{X}_0 \in \mathcal{X}$, where $\Theta$ is compact and $\mathcal{X}_0$ is a nonempty open set.*

*proof:* For our model, for an event $(t_i, m_i)$, its corresponding intensity pattern is chosen by a categorical variable $\boldsymbol{z}(t_i)$,

$$\lambda_{m_i}(t_i \mid \mathcal{H}_{<t_i}) = \sum_{\boldsymbol{z}} p(\boldsymbol{z})\lambda_{m_i}(t_i \mid \mathcal{H}_{<t_i}, \boldsymbol{z}) = \sum_{k=1}^{K} \pi_k \lambda_{m_i}(t_i \mid \mathcal{H}_{<t_i}, \boldsymbol{\theta}^k) \tag{31}$$

Noting the linear relationship between the input and the parameters in the intensity, $\lambda_{m_i}(t_i \mid \boldsymbol{\theta}^k, \mathcal{H}_{<t_i}) = \mu_{m_i}^k + \sum_{u'=1}^{U} \left(\beta_{m_i \leftarrow u'}^k \sum_{n:t_{u'n}<t_i} \kappa(t_i - t_{u'n})\right)$, we denote $x_{u'}(t_i) = \sum_{n:t_{u'n}<t_i} \kappa(t_i - t_{u'n})$, and thus $\boldsymbol{x}(t_i) = (1, x_1(t_i), ..., x_U(t_i))^{\top}$. We then could write the intensity as $\lambda_{m_i}(t_i \mid \boldsymbol{\theta}^k, \mathcal{H}_{<t_i}) = \boldsymbol{x}^{\top}(t_i)\boldsymbol{\theta}^k$, recall $\boldsymbol{\theta}^k = (\mu_{m_i}^k, \boldsymbol{\beta}_{m_i}^k)^{\top}$.

Instead of the original categorical distribution, we could use a corresponding Gumbel-Softmax distribution (Jang et al. (2016)) denoted as $F$, which is a continuous distribution over the simplex $\Delta^{K-1}$ and produce a $k$-dimensional sample vectors $\boldsymbol{y} = (y_1, ..., y_k)^{\top}$ for approximating samples from the categorical distribution with class probabilities $\{\pi_1, ..., \pi_K\}$. Then we could rewrite the mixture of intensities in Eq. (31) as following,

$$\lambda_{m_i}(t_i \mid \mathcal{H}_{<t_i}) = \int_{\Delta^{K-1}} \left(\sum_{k=1}^{K} y_k \lambda_{m_i}(t_i \mid \mathcal{H}_{<t_i}, \boldsymbol{\theta}^k)\right) dF \tag{32}$$

-Due to our model is linear Hawkes process, we have linear relationship between $\boldsymbol{\theta}^k$ and $y_k$,

$$\sum_{k=1}^{K} y_k \lambda_{m_i}(t_i \mid \mathcal{H}_{<t_i}, \boldsymbol{\theta}^k) = \sum_{k=1}^{K} \left(y_k \mu_{m_i}^k + \sum_{u'=1}^{U} y_k \beta_{m_i \leftarrow u'}^k \sum_{n:t_{u'n}<t_i} \kappa(t_i - t_{u'n})\right)$$

we could denote $\tilde{\boldsymbol{\theta}} = (\sum_{k=1}^{K} y_k \mu_{m_i}^k, \sum_{k=1}^{K} y_k \beta_{m_i \leftarrow 1}^k, ..., \sum_{k=1}^{K} y_k \beta_{m_i \leftarrow U}^k)^{\top}$, and thus our original discrete parameter space $\Theta = \{\boldsymbol{\theta}^k\}_{k=1}^{K}$ would be transformed into a continuous space $\tilde{\Theta} \subseteq \mathbb{R}^{U+1}$. We could then denote $g(\boldsymbol{x}^{\top}(t_i)\tilde{\boldsymbol{\theta}}) = \sum_{k=1}^{K} y_k \boldsymbol{x}^{\top}(t_i)\boldsymbol{\theta}^k$ and the corresponding distribution as $F(\tilde{\boldsymbol{\theta}})$,

$$\lambda_{m_i}(t_i \mid \mathcal{H}_{<t_i}) = \int_{\Theta} g(\boldsymbol{x}^{\top}(t_i)\tilde{\boldsymbol{\theta}}) dF(\tilde{\boldsymbol{\theta}}) \tag{33}$$

Then its clear that under this setting we could invoke following two lemmas in Fox et al. (2012).

**Lemma 1.** *Let intensity $\lambda(\cdot)$ be bounded and non-constant and satisfy \*\*$\lambda(0) \neq 0$\*\*. Then the distribution $F^0 \in \mathcal{F}(\tilde{\Theta})$ is uniformly identified over choices of $\tilde{\Theta}$ for the choice $\mathcal{T} = \mathbb{R}^{U+1}$ as the space of $\boldsymbol{x}(t_i)$.*

*proof for Lemma 1:* The skeleton of the proof follows exactly from (Fox et al. (2012)) and we demonstrate this proof works in our setting. We assume the softmax temperature $\tau$ is fixed, and denote the Gumbel-Softmax distribution with true probabilities $\{\pi_1^0, ..., \pi_K^0\}$ as $F^0$. For the purpose of contradiction, we pick a $F^1 \in \mathcal{F}(\tilde{\Theta})$ such that $F^1 \neq F^0$, and we have

$$\int_{\Theta} g(\boldsymbol{x}^{\top}(t_i)\tilde{\boldsymbol{\theta}}) d(F^0(\tilde{\boldsymbol{\theta}}) - F^1(\tilde{\boldsymbol{\theta}})) = 0$$

for all possible $\boldsymbol{x}(t_i) \in \mathbb{R}^{U+1}$.

Let $\sigma$ denote the finite signed measure on $\tilde{\Theta}$ corresponding to $F^0 - F^1$. Then fix $\boldsymbol{\eta} \in \mathbb{R}^{U+1}$, and let $\sigma_{\eta}$ be the finite signed measure on $\mathbb{R}$ induced by the transformation $\tilde{\boldsymbol{\theta}} \mapsto \boldsymbol{\eta}^{\top}\tilde{\boldsymbol{\theta}}$ in the following sense: for all Borel sets of $\mathbb{R}$ we have $\sigma_{\boldsymbol{\eta}}(C) = \sigma\{\tilde{\boldsymbol{\theta}} \in \tilde{\Theta} : \boldsymbol{\eta}^{\top}\tilde{\boldsymbol{\theta}} \in C\}$.

Then at least for all bounded function $f$ on $\mathbb{R}$, $\int_{\tilde{\Theta}} f(\boldsymbol{\eta}^{\top}\tilde{\boldsymbol{\theta}}) d(F^0(\tilde{\boldsymbol{\theta}}) - F^1(\tilde{\boldsymbol{\theta}})) = \int_{\mathbb{R}} f(t) d\sigma_{\boldsymbol{\eta}}(t)$. Therefore by the assumption that our Hawkes process is stationary, our function $g(\cdot)$ is non-constant and bounded, we have,

$$0 = \int_{\tilde{\Theta}} g(\alpha\boldsymbol{\eta}^{\top}\tilde{\boldsymbol{\theta}}) d(F^0(\tilde{\boldsymbol{\theta}}) - F^1(\tilde{\boldsymbol{\theta}})) = \int_{\mathbb{R}} g(\alpha t) d\sigma_{\boldsymbol{\eta}}(t)$$

for all $\alpha \in \mathbb{R}$.

We denote $\boldsymbol{L} = \boldsymbol{L}^1(\mathbb{R})$ for the space of integrable functions on $\mathbb{R}$, and $\boldsymbol{M} = \boldsymbol{M}(\mathbb{R})$ for the space of finite signed measures on $\mathbb{R}$. For $f \in \boldsymbol{L}$, $\hat{f}$ denotes the Fourier transform, and similar we have $\hat{\mu}$ for $\mu \in \boldsymbol{M}$.

First, because we assume $g(0) \neq 0$ and setting $\alpha = 0$ (this could be satisfied if we assume there is another small general base term in all dimensions' intensity), we find that in particular,

$$\int_{\mathbb{R}} d\sigma_{\boldsymbol{\eta}}(t) = \hat{\sigma}_{\boldsymbol{\eta}}(0) = 0 \tag{34}$$

For $\boldsymbol{\eta} = 0$, $\sigma_0$ is concentrated at $t = 0$ and $\sigma_0\{0\} = \hat{\sigma}_0 = 0$, hence $\sigma_0 = 0$.

Now consider $\boldsymbol{\eta} \neq 0$, and the integral

$$\int_{\mathbb{R}} g(\alpha t) d\sigma_{\boldsymbol{\eta}}(t) = 0 \tag{35}$$

Note that now $\sigma_{\boldsymbol{\eta}}$ is absolutely continuous with respect to Lebesgue measure on $\mathbb{R}$ by construction of $\sigma_{\boldsymbol{\eta}}$ from $\sigma$, and thus would have the corresponding Radon-Nikodym derivative $h \in \boldsymbol{L}$. Then $\hat{h} = \hat{\sigma}_{\boldsymbol{\eta}}$ and from above we have $\hat{h}(0) = 0$. Rewriting $\alpha = 1/\tau$ with $\tau \neq 0$ and applying the change of variables $t \mapsto \tau t + s$, we obatain for all nonzero real $\tau$,

$$\int_{\mathbb{R}} g(t + \frac{s}{\tau}) h(\tau t + s) dt = 0 \tag{36}$$

Write $M_\tau h(t)$ for $h(\tau t)$. The above equation implies that $\int_{\mathbb{R}} g(t + c) f(t) dt$ for some c vanishes for all f contained in the closed translation invariant subspace $\boldsymbol{I}$ spanned by the family $M_\tau h$, $\tau \neq 0$, and $\boldsymbol{I}$ is also an ideal in $\boldsymbol{L}$. Following the notation in Rudin(1967), write $Z(f)$ for the set of all $\omega \in \mathbb{R}$ where the Fourier transform $\hat{f}(\omega)$ for $f \in L$ vanishes and define $Z(\boldsymbol{I})$, the zero set of $\boldsymbol{I}$, as the set of $\omega$ where the Fourier transforms of all functions in $\boldsymbol{I}$ vanish.

For the purpose of contradiction, suppose that $h$ is nonzero. As $\hat{M_\tau h}(\omega) = \hat{h}(\omega/\tau)/\tau$ and $\hat{h}(0) = 0$, following the same argument in Hornik(1991) we conclude that $Z(\boldsymbol{I}) = \{0\}$ and also that I is precisely the set of all integrable functions $f$ with $\int_{\mathbb{R}} f(t) dt = \hat{f}(0) = 0$. Because $\boldsymbol{I}$ is an ideal subspace of $\boldsymbol{L}$ and $h$ is nonzero, the above statements together with 36 imply that the integral $\int_{\mathbb{R}} g(t + c) f(t) dt$ for some $c$ vanishes for all integrable functions $f \in \boldsymbol{L}$ that have zero integral. As Hornik(1991) argues, this implies that $g(\cdot)$ must be constant, which was ruled out by our assumption that $g(\cdot)$ is non-constant. Therefore, we conclude $h = 0$ and thus $\hat{h} = \hat{\sigma}_{\boldsymbol{\eta}}$ is identically zero. By the uniqueness Theorem 1.3.7(b) in Rudin (1967), we conclude $\sigma_{\boldsymbol{\eta}} = 0$ for all $\boldsymbol{\eta} \in \mathbb{R}^k$.

To complete the proof, denote the Fourier transform of $\sigma$ at $\boldsymbol{\eta}$ as $\hat{\sigma}(\boldsymbol{\eta}) = \int_{\tilde{\boldsymbol{\Theta}}} \exp(i\boldsymbol{\eta}^\top \tilde{\boldsymbol{\theta}}) d\sigma(\tilde{\boldsymbol{\theta}})$. It follows that,

$$\hat{\sigma}(\boldsymbol{\eta}) = \int_{\tilde{\boldsymbol{\Theta}}} \exp(i\boldsymbol{\eta}^\top \tilde{\boldsymbol{\theta}}) d\sigma(\tilde{\boldsymbol{\theta}}) = \int_{\mathbb{R}} \exp(it) d\sigma_{\boldsymbol{\eta}}(t) = 0$$

and thus $\hat{\sigma} = 0$. Invoking the uniqueness Theorem 1.3.7(b) in Rudin (1967), we conclude $\sigma = 0$ which implies $F^1 = F^0$. This completes the proof for lemma 1. $\blacksquare$

Recall our definition of $x_{u'}(t_i) = \sum_{n:t_{u'n} < t_i} \kappa(t_i - t_{u'n})$ actually implies for any possible $t_i$ and $\mathcal{H}_{<t_i}$ in sample space, we have $\boldsymbol{x}(t_i) \in \mathcal{X} = \mathbb{R}_+^{U+1}$, which means $\mathcal{X}$ is a nonempty open set belong to $\mathbb{R}^{U+1}$. Then noting the following Lemma 2,

**Lemma 2.** *Let $g(\cdot)$ be real analytic and let a set of $x$, $\mathcal{T}$, contain a nonempty open set. The distribution $F^0 \in \mathcal{F}(\tilde{\boldsymbol{\Theta}})$ is uniformly identified over choices of $(\tilde{\boldsymbol{\Theta}}, \mathcal{T}_0)$, with $\tilde{\boldsymbol{\Theta}}$ compact, with nonempty open sets $\mathcal{T}_0 \subset \mathcal{T}$ if and only if $F^0 \in \mathcal{F}(\tilde{\boldsymbol{\Theta}})$ is uniformly identified over compact choices of $\tilde{\boldsymbol{\Theta}}$, for at least one fixed $\mathcal{T}_0 \subseteq \mathcal{T}$.*

Therefore, Lemma 1 states that $F^0$ is identified with $\mathcal{T} = \mathbb{R}^{U+1}$, and Lemma 2 states that $F^0$ is identified with any nonempty open set $\mathcal{T}_0 \subseteq \mathcal{T}$, we then conclude that $F^0$ in our model is uniformly identified in our support $\mathcal{X}$.

# G EXPERIMENT DETAILS

## G.1 SYNTHETIC EXPERIMENT

**Meta-rules:** We mainly consider optimizing following rules in our synthetic experiment. As we deliberately designed our Hawkes process parameters, we could easily note the ground truth preference of dosage for our rules. To be more specific, for rule 1, at state 0 one should choose $A_1$ instead of $A_2$, while for state 1 we should choose $A_2$ instead of $A_1$; for rule 2, at state 0 one should choose $B_1$ instead of $B_2$, while for state 1 we should choose $B_2$ instead of $B_1$.

- **Meta-Rule 1:**
    - **Condition (Fixed)**: If the patient has outcome 1.
    - **Action (Fixed):** Administer Drug $A$.
    - **Learnable Parameters:**
        * Dosage: The specific dosage of Drug $A$, $A_1$ or $A_2$.
        * Timing: The best time to administer the drug, $\tau_A$.
        * Latent State Influence: The influence of a latent state $z$.

- **Meta-Rule 2:**
    - **Condition (Fixed)**: If the patient has outcome 2.
    - **Action (Fixed):** Administer Drug $B$.
    - **Learnable Parameters:**
        * Dosage: The specific dosage of Drug $B$, $B_1$ or $B_2$.
        * Timing: The best time to administer the drug, $\tau_B$.
        * Latent State Influence: The influence of a latent state $z$.

## G.2 REAL-DATA EXPERIMENT

Sepsis is a life-threatening condition that occurs when the body's response to infection causes widespread inflammation, and is a major cause to tissue damage, organ failure, and mortality. Our decision-rule optimization method might be helpful since there is still a great deal of uncertainty regarding clinical recommendations in the management of sepsis (Evans et al. (2021)).

**Dataset description.** MIMIC-III (Johnson et al. (2016)) is a large, publicly available database containing de-identified health data from over 60,000 patients admitted to the ICUs at the Beth Israel Deaconess Medical Center. MIMIC-III is widely used in medical research for developing predictive models, studying disease progression, and analyzing the effects of treatments in critical care settings. It includes detailed information on patient demographics, vital signs, laboratory test results, medications, treatment procedures, and clinical outcomes. The *no unobserved confounders* assumption is typically hard to be satisfied, since treatments and outcomes might be affected by some other factors those are not recorded in the database. Therefore, our method would be appropriate for learning decision-rules in this confounded setting.

**Patients:** We extracted 2000 sequences with the criteria that those patients are diagnosed with sepsis (Saria (2018)) and the corresponding data were not missing. We regard this as our population to apply our EM algorithm for fitting our model. We then select the patients based on our meta-rules and use this subset for our decision-rule optimization process.

**Treatments:** Vasopressor therapy and fluid treatment are used in the management of sepsis with the goal of stabilizing the patient by preserving blood pressure and ensuring proper organ perfusion (Komorowski et al. (2018)). Vasopressors function to constrict blood arteries and boost cardiac output, while fluids aid in restoring intravascular volume. These actions are critical in preventing organ failure and guaranteeing the body receives enough oxygen and nutrients throughout the sepsis response. We list the detailed items in Table 2, with 3 types of fluids, 4 types of vasopressors, and 2 rypes of inotroics.

**Outcomes:**   We treated real-time urine and survival condition as our main outcome indicators, also with several important lab measurements since they also have important impact on urine and survival condition. Low urine output is a critical indicator of kidney dysfunction and often signal the septic chock. Persistent low urine may indicate inadequate response to treatment, ongoing shock, or impending multi-organ failure, making it an essential parameter to monitor in sepsis management. Given the high death rate associated with sepsis, patient survival is a key outcome, and the goal of any treatment is to raise that probability.

**Reward $Y$ design:**   We design our final outcome $Y$ as a deterministic function of all observed outcomes, distinct from the synthetic experiment approach. In this context, since there are no good outcomes, we utilize a squared weighted sum rather than a proportion. Specifically, $Y$ is defined as the square of the weighted sum of the counts of bad outcomes, where each outcome is assigned a dual weighting scheme. The first weight is based on the timing of the event, with later events receiving higher weights to reflect their greater impact. The second weight is determined by the type of outcome, prioritizing severity as follows: $survival$ events are weighted at 0.6, $low - urine$ events at 0.3, and $low - BP$ events at 0.1. Therefore, our target is to find the best decision rules so that $Y$ could be minimized.

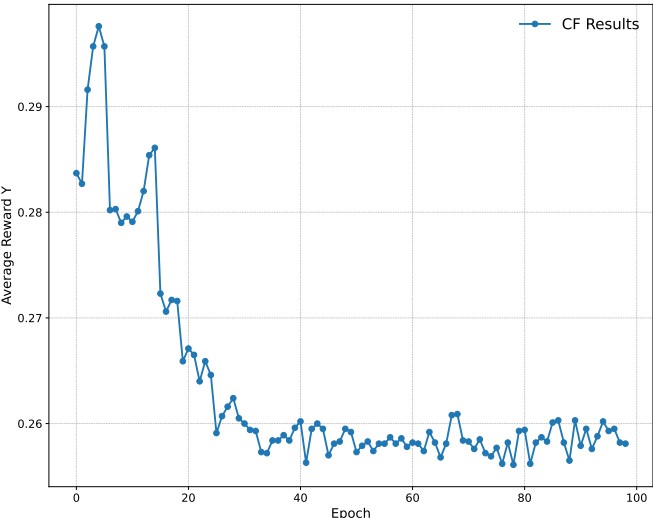

Figure 4: Convergence plot for MIMIC-III.

**Meta-rules:**

- **Meta-Rule 1: Fluids Administration**
    - **Condition (Fixed)**: If the patient shows low urine output.
    - **Action (Fixed):** Administer fluids.
    - **Learnable Parameters:**
        * Type: Colloid, crystalloid, or water.
        * Timing: The best time to administer the fluids, $\tau_f$.
        * Latent State Influence: The influence of a latent state $z$.
- **Meta-Rule 2: Vasopressor Usage**
    - **Condition (Fixed)**: If the patient shows low blood pressure.
    - **Action (Fixed):** Administer vasopressors.
    - **Learnable Parameters:**
        * Type: Norepinephrine, or Dopamine.
        * Timing: The best time to administer the drug, $\tau_v$.
        * Latent State Influence: The influence of a latent State $z$.

Table 2: Treatments and Output Indicators in Real-data Experiment

| Category | Items |
|---|---|
| **Fluid** | Crystalloid |
| | Colloid |
| | Water |
| **Vasopressor** | Epinephrine |
| | Phenylephrine |
| | Norepinephrine |
| | Dopamine |
| **Inotropic** | Dobutamine |
| | Milrinone |
| **Lab Measurement** | Low system blood pressure |
| **Outcome** | Low-urine |
| | Survival |

## H  COMPUTING INFRASTRUCTURE

All synthetic and real-world data experiments were conducted on an Ubuntu 20.04.3 LTS system equipped with an Intel(R) Xeon(R) Gold 6248R CPU @ 3.00GHz and 227 GB of memory.

