# OpenReview forum: "Treatment Rule Optimization Under Counterfactual Temporal Point Processes with Latent States"
_ICLR.cc/2025/Conference — Submitted to ICLR 2025_

### Official Review · Reviewer_VW1G · 2024-10-17

**Soundness:** 2
**Presentation:** 2
**Contribution:** 2
**Rating:** 3
**Confidence:** 3

**Summary:**

The paper proposes a new method for counterfactual prediction and policy learning in a continuous time setting. The authors leverage a probabilistic approach and train their model using a variant of the EM algorithm. The model is evaluated on both simulated and real-world data.

**Strengths:**

- The paper targets an important problem, relevant in many disciplines such as healthcare or economics
- The proposed method is flexible and the probabilistic formulation potentially allows for uncertainty quantification
- The method allows for both counterfactual prediction and policy learning

**Weaknesses:**

- The problem formulation is a little sloppy. For example, in Eq. (7) you use the do operator but never define an appropriate SCM or potential outcomes to work with. Also, notation sometimes seems confusing and redundant, e.g., treatment times and events are first denoted with $t, m_a$ and later with $x, \tau$.
- The related work on counterfactual prediction should be expanded. There are various methods for both discrete and continuous time that have not been cited. Just to list a few: https://papers.nips.cc/paper_files/paper/2018/hash/56e6a93212e4482d99c84a639d254b67-Abstract.html, https://arxiv.org/abs/2204.07258, https://arxiv.org/abs/2002.04083, https://arxiv.org/abs/2310.17463, https://arxiv.org/abs/2407.05287
- It would be helpful if the authors could spill out their contribution as compared to previous work more clearly. In their related work, the authors claim that their method can handle unobserved confounders which is not true for previous work. However, the latent variables $z$ considered in their papers are not confounders (see Fig. 1).
- Related to the above, the novel ideas in this paper seem rather limited. The main contribution seems to be the specification of the latent variable model. The learning algorithm (inner vs outer loop) seems to correspond to a classical policy learning algorithm (where first the policy value is estimated and then the policy is updated) combined with a standard EM algorithm
- In Eq. (7), the authors only condition on the latent states z but not on observed variables (e.g., outcome or treatment history). This does not match with the motivation of the paper (e.g., in the introduction: "Given the observational treatment and outcome trajectories, can we modify specific treatment actions...?")
- The model relies on a variety of parametric assumptions which can restrict flexibility.
- If I understand correctly, the model does not account for (observed) time-varying confounders. These are common in medicine and often available in electronic health records. Taking into account such time-varying confounding necessitates the usage of specialized adjustment mechanisms to correctly estimate counterfactual outcomes/ treatment effects (see here: https://arxiv.org/abs/2407.05287). Ignoring these confounders can severely limit the method's practical applicability.
- The method relies on various assumptions that are not spelled out explicitly, e.g., consistency (no spillover effects), positivity, and ignorability (no unobserved confounders).
- Identifiability: Theorem 1 assumes that the latent variables $z$ are identified in order to identify the model parameters. How can this be ensured in practice? Furthermore, I somewhat doubt the validity of the theoretical arguments as the authors never make use of assumptions necessary to identify counterfactuals or treatment effects (see the point above).

**Questions:**

- Why does the proposed method work with "meta-rules" instead of learning an optimal policy in a fully data-driven manner? It seems like the aim is to "hard-code" certain aspects of the policy. I think this should be justified e.g., with references to relevant literature.

---

> ### Author Response · Authors · 2024-11-25
>
> We sincerely appreciate the time and effort you’ve taken to provide such thoughtful and detailed feedback on our work.
>
> > **Weakness 1:**
>
> - *in Eq. (7) you use the do operator but never define an appropriate SCM or potential outcomes to work with.*
>
> Thank you for your thoughtful feedback. In the updated version, we provided details of the SCM for augmenting the thinning process of MTPP with latent states, and explained the corresponding meaning for our intervention.
>
> - *treatment times and events are first denoted with $t, m_a$ and later with $x, \tau$*
>
> We use the notation $t, m_a$ to represent the treatment events, representing its time and event type. Later we define another notation  $x, \tau$ in the example meta rule, in which $x$ is a potential continuous value of dosage defined in a specific rule, and $\tau$ represent the time-lag for administering this drug once the condition is satisfied. We use this notation since it actually represents the different meaning compared with observed treatment events.
>
> > **Weakness 3: the latent variables $z$ are not confounders (see Fig. 1).**
>
> We appreciate a lot for you concern. We suggest the latent variable $z$ could be regarded as a special kind of latent confounder since we assume the treatment and outcome trajectories are jointly generated by the Ogata’s thinning process for MTPP. Therefore, the latent state $\boldsymbol{z}(t)$ at time $t$ would affect treatment and outcome intensities simultaneously in the acceptance and rejection process for a event, regardless of only one type of the event (roughly belong to treatment or outcome, as shown in fig 1) would be generated.
>
> > **Weakness 5: In Eq. (7), the authors only condition on the latent states z but not on observed variables.**
>
> Thanks a lot for noting this, this was a typo and we've corrected it in the updated version.
>
> > **Weakneess 7: (observed) time-varying confounders.**
>
> We appreciate your valuable suggestion. We suppose when observed time-varying confounders exists, they might be incorporated into the multivariate Hawkes model by assuming them as components of the network. This allows the estimation of their causal relationships, which can then be leveraged during the counterfactual sampling process. However, if time-varying confounders are unobserved, handling them presents a separate challenge, which we plan to address in future work.
>
> > **Weakness 8&9: The method relies on various assumptions that are not spelled out explicitly.**
>
> We are thankful for your comments, in the updated version we provided the detailed assumptions we need in the appendix. Combined with the proof of counterfactual identifiability of the constructed SCM, we are able to guarantee that the counterfactual effects could be identified. In theorem 1, we provide the conditions and proof that the model parameters are identifiable, the unique parameter also plays a crucial role since it guarantees the counterfactual intensity we are interested in is also unique under each revision for the treatment plan.
>
> > **Q: Why does the proposed method work with "meta-rules" instead of learning an optimal policy in a fully data-driven manner? It seems like the aim is to "hard-code" certain aspects of the policy.**
>
> Relying solely on a data-driven policy can be challenging when data is limited, especially in high-stakes areas like healthcare where patient variability and sparse critical events make learning difficult. Meta-rules offer a structured way to embed expert knowledge directly into the policy, which is especially valuable in clinical settings where treatment decisions must comply with established guidelines and safety protocols, and where substantial prior knowledge is available. This idea aligns with the principles of constrained RL for safe exploration,
> - Le, H. M., et al. (2019). Constrained Policy Optimization.

---

> > ### Comment · Reviewer_VW1G · 2024-11-25
> > **Rebuttal acknowledged**
> >
> > I thank the authors for the rebuttal. However, many of my concerns have not sufficiently been addressed, particularly regarding parametric assumptions and novelty. Furthermore, I still have concerns regarding the identifiability of the proposed method, especially since the authors claim that they can deal with unobserved confounders. Responses to the author's points:
> >
> > Weakness 2: It does not seem like the authors have addressed this. In particular, a discussion on how their proposed model adds to the state-of-the-art would be crucial.
> >
> >
> > Weakness 3: If it is the case that the z are indeed latent confounders, Fig. 1 appears to be incorrect. Furthermore, this raises additional questions regarding identifiability (see below).
> >
> > Weakness 7: My concern was not regarding unobserved time-varying confounders, but observed ones. I suppose that simply incorporating these would bias the estimation due to treatment-confounding feedback. Here, adjustment mechanisms specific to the time-series setting must be applied (e.g., G-computation, propensity weighting). As such, I do not believe the proposed method can be extended to time-varying confounders in a straightforward manner.
> >
> > Identifiability: I still have concerns regarding the identifiability results. It is well known in the literature that (point) identification of causal effects under unobserved confounding is generally impossible. The result the authors provide in Sec. 5 only seems possible due to imposing strong assumptions, such as the identifiability of the dimensionality of the latent factors. It is also not clear to me how to interpret the new Assumption 1, or how to ensure that it holds in practice

---

### Official Review · Reviewer_uM1n · 2024-10-22

**Soundness:** 2
**Presentation:** 1
**Contribution:** 1
**Rating:** 3
**Confidence:** 3

**Summary:**

The paper offers a solution for finding optimal treatment strategies in the presense of discrete-time latent states, continuous-time treatment and outcome processes. The paper represents treatment and outcome processes as marked point processes and model them through Hawkes process given past history and latent states. It then treat the latent states and unknowns from models as parameter of interest and apply an EM algorithm to identify.

**Strengths:**

The Hawkes process modeling and the assoicated EM algorithm are intuitive and natural. The paper is contributing into the field of little consideration and can be novel.

**Weaknesses:**

1. The parametric and highly linear modeling of intensity process can be highly questionable. As in a data-rich environment in ICU with great complication and unknown influences across latent states, treatment, and outcome, I found such a modeling untrustworthy. The concern is further aggrevated when one needs to choose the number of latent states by themsselves (and maybe also timing?), in your real-data application. This concern can be somewhat relieved in two ways:
 a. Justify the usage of such modeling in the literature to show this can be a common and acceptable practice, especially in the presence of latent states and continuous-time treatment and outcomes.
 b. In simulation, investigate scenarios under which the modeling assumption (3) is lightly, moderately, and highly violated and investigated  the performance of your framework.
 c. Can you explore the possibility of relaxing modling assumptions, for example, maybe only the latent states part need to be linear while the part involving history of treatment and outcomes can be non-parametric? I believe this is not too much to ask, given that in a similar continuous-time setting, a complete nonparatric identification can be achievable without any parametric assumptions, for example, Continuous-time targeted minimum loss-based estimation of intervention-specific mean outcomes, HC Rytgaard, TA Gerds, MJ van der Laan The Annals of Statistics 50 (5), 2469-2491.

2. We did not observe the latent states. This not only means its value but also the number of latent states and when they happened. The authors seem to be providing this information in the simulation to their estimator. But in practice, how do choose and decide the number of latent states and when they happened? This can be very difficult and unrealistic to assume.
 a. Can you specify how to select the number of latent states adaptively?
 b. Explain how you determined the number and timing of latent states in their real data analysis
 c. Conduct a sensitivity analysis varying the number of latent states

3. It is commonly known that EM algorithm that this paper heavily relies on, suffers from convergence issue and identification issue. Therefore, Theorem 1 is one key ingredient for the paper to be useful. I think this concern about EM algorithm can be relieved by a clear presentation of assumptions in Theorem 1 (instead of citing assumptions from Bonnet et al. (2023). You need an extensive discussion on these assumptions:
 a. When they hold or fails;
 b. How to interpret them both theoretically and in practice (e.g., in your real data application);
 c. Are you able to verify them in your simulations?
 d. Are we able to test them or conduct sensitivity analysis on them?

**Questions:**

Clarify a confusion:
1. One point I am quite doubtful and thus lowering my scoring is. Why did you only model the intensity process for the timing of treatment and outcome but not the mark? In Equation (8) to (9), the likelihood of m_j suddenly disappeared. I found that highly counterintuitive because the dosage of treatment (m_t, j) and disease progression (m_o, j) shall influence the intensity process greatly. Also, their own distribution shall contribute to the likelihood as well.
 a. If marks are not modeled, explain the rationale and discuss potential limitations
 b. Consider extending the model to explicitly incorporate marks in the intensity and likelihood
2. The last point brings up this question, if you did not model the mark of the outcome process, how did you compute the gradient update on Page 6 line 311?
3. Can you clarify what the baseline means in Figure 3? I searched "baseline" in the paper and still is confused.
4. In Step 1: Initialization in Section 6, how to initialize treatment decision rule parameters? Randomly?
5. In Section 3.2, you used x to represent dosage, did you switch to m_{d, k} in Section 6?
6. Page 3 line 151, do you mean z(t) = [z_k]_{k <= t}?

Address a limitation:
1. In the simulation, apart from with latent states but using EM, without latent states by ignoring latent states and apply MLE, one can add a third one by pretending we observe latent states and directly apply your decision rule optimization algorithm in Section 6 (is this "baseline"?)

---

> ### Author Response · Authors · 2024-11-25
>
> We sincerely thank you for your insightful comments and valuable suggestions, which would greatly help us improve our work. In the following responses, equation numbers and line numbers refer to the updated version of the document, while line numbers mentioned in the questions correspond to the original version.
>
> > **Q1&2: Why did you only model the intensity process for the timing of treatment and outcome but not the mark?**
>
>  We do take markers into account in our model, we model treatment and outcome jointly as a multivariate Hawkes model. A multivariate TPP is a special case of marked TPP, in which markers are discrete and a marker would correspond to a dimension. To illustrate, in Eq. (10) and Eq. (11) the intensity would depends on the marker, $\lambda_{m_j}(t_j|\cdot)$, in which $m_j$ represent the dimension.
>
>
> > **Q3&limitation: Can you clarify what the baseline means in Figure 3?**
>
> Here we refer to 'Baseline' as the average reward in the baseline date. From line 438 to line 443, we illustrate the design of our baseline policy for simulating our baseline population data. Our main idea is to design some naive rules so that it would perform relatively not good so that it could be optimized.
>
> > **Q4: In Step 1: Initialization in Section 6, how to initialize treatment decision rule parameters?**
>
> In our experiments, $m_{d, k}$ was initialized to a constant value of 5 and is subsequently transformed using the softmax function. $\tau_{d, k}$ was initialized within a range of random values between 0.5 and 1.
>
> > **Q5: In Section 3.2, you used $x$ to represent dosage, did you switch to $m_{d, k}$ in Section 6?**
>
> Yes. This is mainly due to that in the example rule, we use $x$ to represent a potential continuous value of dosage. However, in our treatment and outcome model, we discretize this continuous value into several markers, e.g., two markers corresponding to low and high dosage. Therefore, in Section 6, we use $m_{d,k}$ instead of the original continuous parameter $x$.
>
> > **Q6: Page 3 line 151, do you mean z(t) = [z_k]_{k <= t}?**
>
> Here $\boldsymbol{z}(t)$ represents the current state at time $t$, which is a $K \times 1$ vector.
>
> > **Weakness1: The parametric and highly linear modeling of intensity process**
>
> We are grateful for your detailed and thoughtful suggestion. The identifiability of the counterfactual effect we aim to evaluate is crucial for our optimization process, thus we employ the parametric model for ensuring this. In the revised version, we also added some references about applications of linear Hawkes in healthcare setting. Thank you for your suggestion about non-parametric setting, which we suppose would be really helpful and feasible for improving our work.
>
> > **Weakness2: How to choose the number of latent states**
>
> Appreciated a lot for you constructive feedback, indeed the assumption for the number of latent states $K$ sometimes might not hold in the real application. We suppose some methods could be employed to address this, for example one could compare the model performance with different state number via cross validation. Also, as in our real experiment, we take the latent state interpretability and domain expertise into consideration. We regard developing models that adapt the number of latent states dynamically as a future work topic.
>
> > **Weakness3: Proof for Theorem 1**
>
> Many thanks for you suggestion, in the updated version we present the assumptions for Hawkes process clearly.

---

> > ### Comment · Reviewer_uM1n · 2024-11-25
> > **Rebuttal acknowledged**
> >
> > I grately thank the author for their explanation towards my questions.
> >
> > However, the weakness part is not handled entirely. The parametric assumption and how to choose number of latent states as it was. Also, I would really appreciate if the author can add more explanation towards assumption 1, as I stated in the first round of comments. The standalone assumption is not providing readers as useful information.

---

> ### Author Response · Authors · 2024-11-28
>
> We sincerely thank you again for your valuable suggestions.
>
> > **Explanation for Assumption 1**
>
> Thank you for you advice, we have updated Section 5 with detailed explanation provided.
>
> > **Potential advantage of parametric assumption and choice for the number of latent states**
>
> While ICU data is often rich and extensive, healthcare scenarios frequently involve limited datasets. This limitation can arise when studying rare diseases, where the patient population is inherently small, or in situations where data access is restricted due to privacy regulations. For instance, detailed data on specific subpopulations or localized cases might not be readily available or shareable. In such contexts, parametric models, which rely on predefined structures and assumptions, are advantageous. These models are better suited for scenarios with limited data because they can leverage domain knowledge and structural constraints to produce reliable and interpretable results even when data is relatively sparse. Therefore, we still believe our model would be applicable in many practical setting. Also, the ability to provide interpretable parameters for professionals is crucial and desirable under many cases, and allowing professionals to incorporate domain knowledge easily, such as assumptions about biological distributions or event processes, aligns closely with established medical expertise, improving model reliability in sparse data environments.
>
> Also, given the abundance of prior knowledge in healthcare, we believe that in many practical cases determining the number of latent states can be straightforward — for instance, by leveraging well-defined stages or progressions of a disease, or by summarizing static indicators to represent patient conditions effectively. Therefore, we believe our methods will remain highly beneficial and practical across many potentially concerned healthcare scenarios by doctors.

---

### Official Review · Reviewer_y7mL · 2024-10-27

**Soundness:** 3
**Presentation:** 3
**Contribution:** 3
**Rating:** 5
**Confidence:** 3

**Summary:**

The paper proposes a counterfactual treatment optimization framework based on temporal point processes for evaluating and optimizing treatment decision rules in high-stakes areas like healthcare. The framework consists of an outer loop for exploring and optimizing treatment decision rules, and an inner loop for evaluating these rules via counterfactual sampling of symptom events. To address challenges posed by latent states, a two-stage procedure is introduced - first inferring latent states and associated noise to mitigate biases, and then conducting counterfactual sampling. Identifiability of model parameters in the presence of latent states is theoretically proven, enhancing robustness. The framework aims to refine existing treatments and generate insights for optimizing patient outcomes.

**Strengths:**

1. Tackling a crucial problem of treatment optimization under latent confounding in a counterfactual manner.

2. Integrating temporal point processes to model event sequences rigorously.

3. The paper establishes strong theoretical guarantees through comprehensive identifiability proofs. This mathematical rigor ensures the reliability of counterfactual evaluations and strengthens the practical applicability of their approach.

**Weaknesses:**

1. The notation of the do-operator seems a bit confusing. For example, on line 212, it would be helpful if the meaning of $do(H_a(T) = H'_a(T')|{f_1, \ldots, f_D}, {z_1, \ldots, z_K})$ could be mathematically illustrated.

2. The authors did not compare their approach with other off-policy optimization methods in the experiments, which might raise some doubts about the effectiveness of their algorithm.

3. While the paper provides detailed reward optimization results for synthetic experiments (Section 7.1), presenting similar quantitative analysis for the MIMIC-III database (e.g., convergence plots and reward comparisons similar to Figure 3) would help readers better appreciate how the proposed framework translates to real-world healthcare scenarios.

**Questions:**

1. What if the number of latent factors K is unknown, or the latent states are not categorical - will the identifiability result still establish under such relaxed assumptions?
2. How can the framework handle scenarios where meta-rules are not well-defined or unavailable?

---

> ### Author Response · Authors · 2024-11-25
>
> We appreciate a lot for your constructive feedback.
>
> > **Q1: What if the number of latent factors K is unknown, or the latent states are not categorical - will the identifiability result still establish under such relaxed assumptions?**
>
> Our current proof technique requires knowledge of $K$, but this proof might be adaptable to allow for latent variables that are not categorical, which would, however, require further careful consideration.
>
> > **Q2: How can the framework handle scenarios where meta-rules are not well-defined or unavailable?**
>
> Since our framework operates within a counterfactual scenario, our focus is on revising or adding treatment events to the observed treatment sequence based on the predefined meta-rules. Typically, the observed treatment sequence reflects some rational plans established by doctors. Therefore, if our meta-rules are poorly defined, the counterfactual outcome may worsen, signaling that our meta-rules might need adjustment.
>
> While an initial guess about certain meta-rules may be necessary, feedback from the framework as we mentioned above can indicate whether these self-defined rules are appropriate, allowing for subsequent adjustments. Additionally, one might consult professionals or use AI assistance to receive guidance on suggested rules in their setting.
>
> > **Weakness1: notation of the do-operator.**
>
> In the updated version, we provided detailed construction of the SCM and a brief further explanation about do-operator.
>
> > **Weakenss2**
>
> Our approach introduces a novel adaptation of a counterfactual framework to multivariate TPPs, with a unique focus on optimizing meta-rules under this counterfactual setting. To the best of our knowledge, we are the only one address this combination. We thus suppose performing a fair comparison requires methods that also operate within a counterfactual framework tailored for TPPs. However, most off-policy evaluation or optimization techniques do not incorporate counterfactual reasoning. This highlights both the distinctiveness of our work and the challenges of benchmarking against existing methods.
>
> > **Weakness3**
>
> Due to the page length limitation, we add this plot in the appendix.

---

### Official Review · Reviewer_sRX6 · 2024-11-04

**Soundness:** 1
**Presentation:** 1
**Contribution:** 1
**Rating:** 3
**Confidence:** 2

**Summary:**

The paper develops a new method for time-varying off-policy learning based on confounded irregularly-sampled data. The authors assume observational data is generated by a Hawkes process with discrete latent variables, which capture hidden confounding. Then, given the observational data, they (1) employ an expectation-maximisation (EM) algorithm to infer the latent variables, (2) generate counterfactual outcomes, and (3) perform gradient-based policy learning. The method was evaluated based on one synthetic and one real-world experiment.

**Strengths:**

The paper aims to tackle a challenging problem of causal inference and reinforcement learning: time-varying off-policy learning based on the irregularly sampled, confounded observational data and there are not that many related methods tailored for this setting.

**Weaknesses:**

I find it very hard to acknowledge the contribution of the paper due to several major flaws:
1. It was unclear, what causal assumptions the paper relies on. Specifically, does one need to assume a non-Markovian time and time-varying potential outcomes framework [1],  or some Markovian decision process (MDP)? Also, additional identifiability assumptions are required due to the presence of hidden confounding [2-3]. I am not even sure, whether the paper works with the interventional quantities (e.g., potential outcomes after the interventions in the future), or counterfactual quantities (e.g., the outcomes after intervention on the past treatments). The latter one, for example, requires even stronger assumptions [4] (e.g., invertibility of the latent noise of the outcome).  Therefore, it is hard to judge whether the proposed method is sound or even correct.
2. The authors did not provide any derivations for the identification of the target causal quantity, i.e., policy value after the intervention (Theorem 1 relates to the identifiability of the latent variables, which is not the same as policy value identification).
3. Lack of baselines. Although a lot of relevant methods are mentioned in the related works section, none are provided in the synthetic benchmark.
4. Poor quality. Multiple notions and notations are not properly introduced or defined throughout the paper. For example, $\phi(t-s)$, (Eq. 3); triggering functions; $x$, (line 208); etc. Thus, it is hard to understand the theoretical claims of the paper.

As all the above-mentioned problems cannot be easily fixed during the rebuttal, I recommend rejecting this work.

  References:
- [1] Robins, J. M. and Hern  ́an, M. A. Estimation of the causal effects of time-varying exposures. CRC Press, Boca Raton, FL, 2009.
- [2] Milan Kuzmanovic, Tobias Hatt, and Stefan Feuerriegel. Deconfounding temporal autoencoder: estimating treatment effects over time using noisy proxies. In Machine Learning for Health, pp. 143–155. PMLR, 2021.
- [3] Ioana Bica, Ahmed Alaa, and Mihaela Van Der Schaar. Time series deconfounder: Estimating treatment effects over time in the presence of hidden confounders. In International conference on machine learning, pp. 884–895. PMLR, 2020.
- [4] Hızlı, Çağlar, et al. "Causal Modeling of Policy Interventions From Sequences of Treatments and Outcomes." arXiv preprint arXiv:2209.04142 (2022).

**Questions:**

- Why are $H_0$ and $H_1$ two distinct nodes on the causal diagram in Fig. 1? Shouldn’t $H_0$ be a part of $H_1$?
-  What mixture distribution is meant in line 152? Isn’t a mixture of categorical variables also a categorical variable?
- What is the intuition of the counterfactual interventions in the past (namely both conditioning and intervening on the same treatments) from the practical point of view? Should regular future interventions be sufficient for developing optimal treatment regimes?

---

> ### Author Response · Authors · 2024-11-25
>
> Thank you so much for your questions and helpful suggestions. In the following responses, equation numbers and line numbers refer to the updated version of the document, while line numbers mentioned in the questions correspond to the original version.
>
> > **Q1: why are $H_0$ and $H_1$ two distinct nodes**
>
> $H_0$ is part of $H_1$, thus there is a rightarrow from $H_0$ to $H_1$.
>
> > **Q2: What mixture distribution is meant in line 152?**
>
> Thank you for your detailed feedback, this was a typo and we have corrected that.
>
> > **Q3: What is the intuition of the counterfactual interventions in the past  from the practical point of view? Should regular future interventions be sufficient for developing optimal treatment regimes?**
>
> While future interventions are essential for developing optimal treatment regimes, counterfactual interventions in the past add a crucial layer of insight. To illustrate, in high-stakes settings like healthcare, we cannot go back in time to test whether an alternative treatment plan would have benefited a patient who did not survive. Counterfactual analysis addresses this by focusing on posterior noise given observed information, rather than the prior noise used in interventional analysis in a SCM. One could imagine that this approach allows us to simulate conditions that closely resemble the environment at the time the data was observed, enabling more rigorous assessments of alternative treatment plans, e.g., as mentioned in the following two references,
> - Oberst, M., & Sontag, D. (2019). Counterfactual Off-Policy Evaluation with Gumbel-Max Structural Causal Models;
> - Buesing, L., Weber, T., Zwols, Y., Racaniere, S., Guez, A., Lespiau, J. B., & Heess, N. (2018). Woulda, coulda, shoulda: Counterfactually-guided policy search.
>
> > **Weakness 1 & 2**
>
> Many thanks for your valuable feedback. We model treatment and outcome events jointly via a multivariate temporal point process. We aim to evaluate and optimize a counterfactual quantity by revising the treatment trajectory based on meta-rules. These revisions would influence the intensity of outcome dimensions, enabling us to define a counterfactual intensity of interest. To analyze this, we construct a SCM to augment the thinning process for MTPP. Using this framework, we could sample counterfactual outcomes, and the identifiability of these counterfactuals is ensured by the properties of the constructed SCM. We've uploaded a revised version, provided the causal assumptions we need and the counterfactual identifiability proof, so that our target counterfactual quantity is identified.
>
> > **Weakness 4**
>
> We suppose we provided the detailed definition for each notation, for example,
>
> - $\phi (t-s)$ in Eq. (3) -- explained in line 173.
>
> - triggering functions -- defined in line 180 and Eq. (4).
>
> - $x$ in line 208 (original ver) - defined in example meta-rule , line 250.

---

> > ### Comment · Reviewer_sRX6 · 2024-11-26
> >
> > Thank you for your rebuttal!
> >
> > I want to elaborate further on my questions and the aforementioned weaknesses.
> >
> > Q1. According to a general definition of a causal diagram, every node should be (in principle) independently intervenable [1]. Therefore, $H_0$ and $H_1$ cannot be simultaneously two different nodes and a part of one another.
> >
> > Q3. Can you provide real-world examples of when this counterfactual off-policy learning can be applied?
> >
> > W1. What do you mean by "These revisions would influence the intensity of outcome dimensions"?
> >
> > W2. So do I understand it right that you make a very strong assumption of a specific SCM?
> >
> > W3. I think the authors did not answer this weakness.
> >
> > W4. The definitions should be introduced immediately before or after some notation is used. The paper is thus very hard to read due to the cross-references.
> >
> > Considering the unanswered weaknesses and the remaining concerns, I decided to remain my score.
> >
> > References:
> > - [1] Pearl, Judea. "Models, reasoning and inference." Cambridge, UK: CambridgeUniversityPress 19.2 (2000): 3.

---

> > > ### Author Response · Authors · 2024-11-28
> > >
> > > Thank you again for your constructive feedback!
> > >
> > > > **Q1**
> > >
> > > We design this figure by regarding Figure 1 in [1] as the main reference, in which Buesing et. al also use node $H_t$ to represent a history node, and $H_{t-1}$ is part or $H_t$. Thank you for pointing out this, due to page limitation and to avoid confusion we moved figure 1 from main part.
> > >
> > > [1] Buesing, L., Weber, T., Zwols, Y., Racaniere, S., Guez, A., Lespiau, J. B., & Heess, N. (2018). Woulda, coulda, shoulda: Counterfactually-guided policy search.
> > >
> > > > **Q3**
> > >
> > > We suppose this method would be valuable in many high-stake settings, where testing alternative interventions retrospectively is impossible. For instance, consider a scenario involving rare and severe diseases where clinical trials are infeasible due to limited patient populations or ethical constraints. Suppose a patient receives a particular treatment but experiences a poor outcome; we cannot rewind time to test whether a different treatment plan might have improved their condition. Counterfactual analysis allows us to infer the potential effects of alternative treatments using observed data, then refine our interested treatment strategies.
> > >
> > > > **Weakness1**
> > >
> > > From Equation (3) for outcome intensities $\boldsymbol{\lambda_o^*(t|\cdot)}$, we notice that there is a trigerring effect from treatment dimensions to outcome, $\int_0^t  \boldsymbol{\phi_{o \gets a}} (t-s) d \boldsymbol{N_a}(s)$. Therefore, if we revise the treatment plan from $H_a(T)$ to $H_a^{'}(T)$, the value of this integral would change, then result in a revised intensity for outcome dimensions, $\boldsymbol{\lambda}_{{\rm cf}}$,  that we are interested in.
> > >
> > > > **Weakness2**
> > >
> > > Yes. This special kind of SCM is constructed for multivariate TPP and assumptions are required to guarantee counterfactual identifiability, similar ideas as in previous works,
> > >
> > > - Noorbakhsh, Kimia, and Manuel Rodriguez. "Counterfactual temporal point processes." Advances in Neural Information Processing Systems 35 (2022): 24810-24823.
> > >
> > > - Hızlı, Çağlar, et al. "Causal Modeling of Policy Interventions From Sequences of Treatments and Outcomes." arXiv preprint arXiv:2209.04142 (2022).
> > >
> > > > **Weakness3**
> > >
> > > We performed counterfactual optimization within the framework of multivariate TPPs, where methods remain relatively unexplored. The methods mentioned in the related works mainly focus on the counterfactual evalution only, and under a relatively different model framework. Due to this, we could hardly find comparable approaches currently to conduct reasonable comparisons. This absence of precedent underscores our contribution, and we acknowledge the value of future work in establishing benchmarks as similar methodologies emerge.
> > >
> > > > **Weakness4**
> > >
> > > Thank you for your kindly suggestion, we've checked the notations again and ensured each notation was introduced immediately before or after we use it.

---

> > > > ### Comment · Reviewer_sRX6 · 2024-12-02
> > > >
> > > > Thank you for the new answers!
> > > >
> > > > Regarding Q3, I still don't understand why policies based on the interventional quantities wouldn't suffice in this case.

---

> > > > > ### Author Response · Authors · 2024-12-04
> > > > >
> > > > > Thank you for your further question. We suppose that in previous work they suggest counterfactual reasoning provides a more rigorous framework for designing better targeted interventions. For example, in [1] they highlights that counterfactual reasoning helps humans refine and adjust their strategies after unsuccessful outcomes by considering hypothetical alternatives, which suggest counterfactual reasoning can help humans in designing more effective targeted interventions based on the outcomes of past interventions.
> > > > >
> > > > > [1] Epstude, K., & Roese, N. J. (2008). The functional theory of counterfactual thinking. Personality and social psychology review, 12(2), 168-192.

---

### Meta-Review · Area_Chair_txxu · 2024-12-10

**Metareview:**

Optimizing sequential policies is of much importance, and definitely a major AISTATS topic. The continuous-time modeling and counterfactual sampling show potential, but reviewers and myself continued to struggle in appreciating the novel aspects and the fundamental needs of counterfactual sampling. Despite the references given, following the finer details wasn't all that clear - for instance, counterfactual sampling requires some sort of cross-world inference, while more standard approaches for causal RL (including Q-learning with causal assumptions) wouldn't require that level of untestable modeling. Maybe there are clearer ways of presenting it, but currently even equations like (9) use non-standard notation like $p(z | do(x | y))$ (if I understood it correctly, the intention would be $p(z |y, do(X = f(y)))$? I wasn't sure)

A promising set of ideas, but we concluded that the manuscript will have much to gain by another round of preparation.

**Additional Comments On Reviewer Discussion:**

Discussion was comparatively light, but I read carefully the threads between authors and reviewers. I think there was progress, but a fresh round of writing will benefit the paper, its audience, and ultimately the authors.

---

### Decision · Program_Chairs · 2025-01-22

Reject